# Mechanism of CO$_2$ and NH$_3$ transport through human aquaporin 1: Evidence for parallel CO$_2$ pathways

Raif Musa-Aziz[1,2] , R. Ryan Geyer[2], Seong-Ki Lee[2] , Fraser J. Moss[2] and Walter F. Boron[2,3,4]

[1]*Department of Physiology and Biophysics, Institute of Biomedical Sciences, University of São Paulo, São Paulo, Brazil*

[2]*Department of Physiology and Biophysics Case Western Reserve University School of Medicine, Cleveland, OH, USA*

[3]*Department of Medicine, Case Western Reserve University School of Medicine, Cleveland, OH, USA*

[4]*Department of Biochemistry, Case Western Reserve University School of Medicine, Cleveland, OH, USA*

Handling Editors: Kim Barrett & Peying Fong

The peer review history is available in the Supporting Information section of this article (https://doi.org/10.1113/JP289556#support-information-section).

**Abstract figure legend** The figure illustrates the permeation pathways for CO$_2$, NH$_3$ and H$_2$O through AQP1. The panel on the left, with the wild-type protein under control conditions, CO$_2$ moves through monomeric pores (2 of 4 shown) but predominantly via an alternate pathway (perhaps the central pore of the tetramer). The NH$_3$ and H$_2$O diffuse exclusively via monomeric pores. The panel on the right, with the use of inhibitors and a point mutation, illustrates the dissection of the two distinct pathways that underlie the selectivity of AQP1 for its molecular traffic.

**Raif Musa-Aziz** earned her PhD in Human Physiology at the University of São Paulo (USP) under Professors Margarida de Mello-Aires and Gerhard Malnic. She completed postdoctoral training with Dr. Walter Boron at Yale and Case Western Reserve Universities, where she developed a method to measure surface pH in *Xenopus laevis* oocytes, advancing studies of CO$_2$ and NH$_3$ transport through membrane proteins. Since 2009, she has been an Assistant Professor at the Physiology & Biophysics Department, Biomedical Sciences Institute-USP. Due to Brazilian *Xenopus* restrictions, her ongoing research focuses on NH$_3$ and H$_2$O transport via urea transporters expressed in *Lithobates catesbeianus* oocytes.

R. Musa-Aziz and R. R. Geyer contributed equally to this work.

This article was first published as a preprint. Musa-Aziz R, Ryan Geyer R, Moss FJ, Boron WF. 2025. Mechanism of CO$_2$ and NH$_3$ Transport through Human Aquaporin 1: Evidence for Parallel CO$_2$ Pathways. bioRxiv. https://doi.org/10.1101/2025.02.28.640247

**Abstract** The traditional view had been that dissolved gases cross membranes simply by dissolving in and diffusing through the membrane lipid. However, some membranes are impermeable to CO$_2$ and NH$_3$, whereas some aquaporin (AQP) water channels—tetramers with hydrophobic central pores—are permeable to CO$_2$, NH$_3$ or both. Nevertheless, we understand neither the routes that CO$_2$ and NH$_3$ take through AQP tetramers, nor the basis of CO$_2$/NH$_3$ selectivity. Here, we show—for human AQP1 (hAQP1)—that virtually all NH$_3$ and H$_2$O pass through the hydrophilic, monomeric pores. However, CO$_2$ passes through both the monomeric pores and another pathway. We expressed hAQP1 in *Xenopus* oocytes and used microelectrodes to monitor the maximal surface-pH transient ($\Delta$pH$_S$) caused by CO$_2$ or NH$_3$ influxes. We found that p-chloromercuribenzene sulfonate (pCMBS)—which reacts with C189 in the monomeric pore—eliminates the entire hAQP1-dependent (*) NH$_3$ signal ($\Delta$pH$_S$*)$_{NH3}$, but only half of the signals for CO$_2$ ($\Delta$pH$_S$*)$_{CO2}$ or osmotic water permeability $P_f$*. 4,4'-diisothiocyanatostilbene-2,2'-disulfonate (DIDS), eliminates the remaining ($\Delta$pH$_S$*)$_{CO2}$ but has no effect on ($\Delta$pH$_S$*)$_{NH3}$ or $P_f$*. Together, the two drugs completely eliminate the CO$_2$ permeability of hAQP1. When we express hAQP1 in *Pichia pastoris*, treat spheroplasts with DIDS and examine hAQP1 by SDS-PAGE, reactivity with an anti-DIDS antibody shows that DIDS crosslinks hAQP1 monomers. Our results provide the first evidence that a molecule can move through an AQP via a route other than the monomeric pore, and raise the possibility that selectivity depends on the extent to which CO$_2$/NH$_3$ moves through monomeric pores versus an alternate pathway (e.g., the central pore).

(Received 23 June 2025; accepted after revision 10 November 2025; first published online 6 December 2025)

**Corresponding authors** R. Musa-Aziz: Department of Physiology and Biophysics, Institute of Biomedical Sciences, University of São Paulo. São Paulo, SP Brazil. Email: raifaziz@icb.usp.br; W. F. Boron: Department of Physiology and Biophysics Case Western Reserve University School of Medicine, Cleveland, OH, USA. Email: walter.boron@case.edu

**Key points**

- Some membranes have negligible CO$_2$ permeability in the absence of protein channels like aquaporin-1 (AQP1).
- We confirm that, during CO$_2$ influx, heterologous expression of human AQP1 (hAQP1) in *Xenopus* oocytes increases the magnitude of the transient surface-pH increase by an amount ($\Delta$pH$_S$*)$_{CO2}$, measured with microelectrodes. During NH$_3$ influx, hAQP1 expression increases the magnitude of the transient pH$_S$ decrease by ($\Delta$pH$_S$*)$_{NH3}$.
- p-chloromercuribenzene sulfonate (pCMBS), which reacts with C189 in the monomeric pore, reduces ($\Delta$pH$_S$*)$_{CO2}$ by half; ($\Delta$pH$_S$*)$_{NH3}$, to zero; and AQP1-dependent osmotic water permeability ($P_f$*), by half.
- 4,4'-diisothiocyanatostilbene-2,2'-disulfonate (DIDS) reduces ($\Delta$pH$_S$*)$_{CO2}$ by half, but has no effect on ($\Delta$pH$_S$*)$_{NH3}$ or $P_f$*. DIDS crosslinks AQP1 monomers expressed in *Pichia pastoris*.
- Together, pCMBS+DIDS reduce ($\Delta$pH$_S$*)$_{CO2}$ to zero. The C189S mutation of AQP1 eliminates the effects of pCMBS, but not of DIDS. Our results thus show that CO$_2$ traverses AQP1 via the monomeric pore plus a novel DIDS-sensitive route that may be the central pore.

# Introduction

Before the discovery of specific protein-mediated pathways in cell membranes, investigators had believed that many small molecules (e.g., H$_2$O, urea, glycerol, lactic acid)—including CO$_2$—freely permeated all membranes to the extent that they can dissolve in and then diffuse through the lipid phase. Solubility-diffusion theory (S-DT) embodies this concept (reviewed in Boron, 2010;

Michenkova et al., 2021). For molecules other than dissolved gases, the generalisability of this notion faded with the discovery of each new membrane channel or transporter, consistent with the idea that cells evolved to achieve tight control over the transmembrane traffic of small molecules. We now believe that the traffic of such molecules depends on some combination of membrane proteins and S-DT (responsible for non-specific leaks).

Regarding $CO_2$ traffic across biological membranes, the first inconsistency in the applicability of S-DT was the demonstration that apical membranes (i.e., facing the lumen) of single gastric glands are impermeable to $CO_2$ (Waisbren et al., 1994). The second was the discovery that the human (h) aquaporin-1 (AQP1), besides its eponymous substrate, $H_2O$ (Preston & Agre, 1991), also conducts $CO_2$ (Nakhoul et al., 1998). Soon after Nakhoul's observation, Cooper and Boron (1998) found that pCMBS inhibits $CO_2$ permeability in oocytes expressing wild-type (WT) hAQP1. Prasad et al. (1998) then reported that AQP1 purified from human red blood cells (RBCs) and reconstituted into lipids from *E. coli* increases $CO_2$ permeability, an effect blocked by $HgCl_2$. Moreover, Forster et al. (1998) found that 4,4'-diisothiocyanatostilbene-2,2'-disulfonate (DIDS) applied to native human RBCs reduces not only $HCO_3^-$ permeability, but also $CO_2$ permeability.

Forster and colleagues recognised that an explanation for their data could be that DIDS blocks a protein pathway for $CO_2$ diffusion through the RBC membrane. Indeed, Endeward et al. (2006) later found that AQP1 is responsible for about half of the $CO_2$ traffic across the human RBC membrane, with the rhesus (Rh) complex being responsible for most of the rest (Endeward et al., 2008), leaving <10% to be accounted for by S-DT. Uehlein et al. (2003) reported that NtAQP1 in tobacco plants plays important physiological roles in $CO_2$ uptake—driven by a very small air-to-chloroplast gradient—for photosynthesis, stomatal opening and leaf growth. Wang and colleagues demonstrated that the $CO_2$ permeability of tomato aquaporin PIP2;1 is essential for $CO_2$-dependent regulation of stomatal aperture (Wang et al., 2016).

Besides $H_2O$ and $CO_2$, various AQPs can conduct a variety of substances, including glycerol in the case of members of the aquaglyceroporin subfamily (Gomes et al., 2009). Yasui et al. (1999) found that AQP6, mainly present in intracellular vesicles, has negligible $H_2O$ permeability but behaves as a non-selective anion channel. Moreover, a single mutation, N60G, abolishes anion permeability and effectively converts AQP6 to a water channel. Later, Qin and Boron (2013) found that the opposite mutation in hAQP5, G50N, eliminates both $H_2O$ and $CO_2$ permeability, whereas the nearby mutation L51R converts AQP5 into an anion channel blockable by mercury. These residues, although on TM2, are in proximity to the ar/R selectivity filter of the monomeric pore. Yool et al. (1996) reported that, with oocytes expressing hAQP1, forskolin or a cAMP analog increased water permeability and triggered a cation conductance. However, others report that they have not been able to confirm that work (Agre et al., 1997; Deen et al., 1997; Verkman & Yang, 1997).

Regarding dissolved gases, hAQP1 conducts $NH_3$ (Nakhoul et al., 2001) and NO (Herrera et al., 2006).

Moreover, different AQPs can exhibit strong selectivity for $CO_2$ over $NH_3$ or vice versa (Geyer et al., 2013; Musa-Aziz, Chen et al., 2009). More recent work has suggested that *Nicotiana tabacum* PIP1;3 (Zwiazek et al., 2017). Pre-print publications have concluded that AQP1, the Rh complex and an unidentified protein are responsible for nearly all $O_2$ permeability in murine RBCs (Moss et al., 2025; Occhipinti et al., 2025; Zhao et al., 2025). Moreover, Al-Samir et al. (2025) report that hAQP1 is permeable to $O_2$. These results are consistent with the idea that cells, by regulating channel expression and trafficking to specific membranes, could control the permeability to $CO_2$, $O_2$ and $NH_3$.

Structural studies show that AQP1 is a homotetramer with four independent monomeric pores—lined mainly by hydrophilic residues—that conduct $H_2O$ (Sui et al., 2001; Walz et al., 1997). At the core of its square-shaped array of four monomers, AQP1 has a central pore—lined exclusively by hydrophobic residues—with no established function. Each monomer spans the lipid bilayer six times and has cytoplasmic amino and carboxyl termini. As predicted by the 'hourglass model' of Preston and Agre (1991), two consensus NPA motifs—on intracellular and extracellular loops that dip in towards the middle of AQP1, near the plane of the membrane—contribute importantly to the monomeric pore. The well-established reduction of osmotic water permeability ($P_f$) by $HgCl_2$ or pCMBS occurs as these agents react with cysteine-189 (C189), two residues upstream from the second NPA (Preston et al., 1993). Replacing this cysteine with serine (C189S), which lacks a sulfhydryl group, eliminates this inhibition (Preston et al., 1993).

We previously introduced an approach for using blunt, pH-sensitive microelectrodes to monitor transient changes in extracellular-surface pH ($pH_S$), caused by the entry of $CO_2$ or $NH_3$ into *Xenopus* oocytes heterologously expressing membrane proteins (Endeward et al., 2006; Musa-Aziz, Jiang et al., 2009, Musa-Aziz, Chen et al., 2009, 2010, 2014a). Mathematical modelling has elucidated the interpretation of $CO_2$-dependent pH transients (Calvetti et al., 2020; Occhipinti et al., 2014; Somersalo et al., 2012). Using this pH approach in a survey of mammalian AQPs, we found that AQPs 1, 6 and 9 are permeable to both $CO_2$ (exposure to 5% $CO_2$) and $NH_3$ (0.5 mM $NH_4Cl$); AQP0, the M23 variant of AQP4 (AQP4-M23) and AQP5 are permeable to $CO_2$ but not $NH_3$; AQPs 3, 7 and 8 are permeable to $NH_3$ but not $CO_2$; and AQP2 as well as AQP4-M1 appear to be permeable to neither (Geyer et al., 2013; Musa-Aziz, Chen et al., 2009). Using an osmotic-shrinkage assay with an 80-fold higher extracellular [$NH_3$] than in our studies, Assentoft et al. detected $NH_3$ permeability through AQP4-M23 (Assentoft et al., 2016). A key unanswered question is how various AQPs exhibit such strikingly different $CO_2/NH_3$ selectivities. The answer presumably

lies in the pathways that these solutes take through various AQP tetramers. However, despite insights from molecular-dynamics simulations, we still lack physiological evidence that establishes the pathways by which either CO$_2$ (Hub & de Groot, 2006; Wang et al., 2007) or NH$_3$ (Kirscht et al., 2016) moves through any AQP. In the case of AtTIP2;1 from the plant *Arabidopsis thaliana*, the crystal structure reveals a unique selectivity filter with a side pore. Kirscht et al. (2016) suggest that NH$_4^+$ could enter from the vacuolar surface (topologically equivalent to the extracellular side), followed by deprotonation to NH$_3$+H$^+$. The NH$_3$ would move along the monomeric pore, whereas the H$^+$ would recycle via the side pore to the vacuolar space.

The purpose of the present study is to explore the pathway(s) by which CO$_2$ and NH$_3$ diffuse through human AQP1. Our approach was to use a combination of cell physiology and biochemistry. Heterologously expressing hAQP1 or its C189S mutant in *Xenopus* oocytes, and using the pH$_S$ approach, we assessed the effects of pCMBS and DIDS on CO$_2$ and NH$_3$ permeation. We also assessed $P_f$ so that we could normalise CO$_2$ and NH$_3$ data to H$_2$O permeability. We examine pCMBS because mercury—acting on monomeric pores—reduces both the $P_f$ (Preston et al., 1993) and CO$_2$ permeability of hAQP1 (Cooper & Boron, 1998), as expressed heterologously in oocytes. We work with DIDS because this drug reduces CO$_2$ permeability in RBCs and in oocytes expressing hAQP1 (Endeward et al., 2006). We find that virtually all of the hydrophilic NH$_3$ molecules—as well as H$_2$O, as expected—pass through the four hAQP1 monomeric pores, which each possess three hydrophilic nodes in their selectivity filters that are located at the centre of an otherwise long hydrophobic channel (Sui et al., 2001). On the other hand, the more hydrophobic CO$_2$ travels both via this pCMBS-sensitive pathway—the only established pathway through any AQP— and as an independent pathway that we can block with DIDS. Studies on hAQP1 expressed in *Pichia pastoris* indicate that DIDS crosslinks hAQP1 monomers. Our work is the first to demonstrate that a substance can move through an AQP via a pathway—possibly the hydrophobic central pore—that is distinct from the monomeric pore. Considering that mutations in human AQPs are associated with a wide variety of pathological conditions (see Verkman, 2008; Verkman et al., 2008), our work could lead to new insights into disease mechanisms, diagnostic tools and therapeutic approaches for improving clinical outcomes.

## Methods

For previous summaries of our approach in oocyte experiments, see Musa-Aziz et al. (2010), Musa-Aziz,

Chen et al. (2009), Musa-Aziz, Jiang et al. (2009). The following description is in greater depth.

### Ethical approval and animal procedures

The Institutional Animal Care and Use Committee at Case Western Reserve University approved the protocols for housing and handling of *Xenopus laevis*, used as a source of oocytes, and rabbits for generating polyclonal antibodies [PHS Assurance number - D16-00089 (A3145-01)].

**Frogs.** We purchased adult female *Xenopus laevis* frogs (NASCO Inc., Fort Atkinson, WI, USA) and housed them in a 20-gallon static aquarium, managed by the Animal Resources Centre (ARC) of the School of Medicine. For stress mitigation, the tank had six or fewer frogs, and for environmental enrichment, we included a PVC elbow pipe. A charcoal Bio-Bag aquarium power pump (Tetra, Blacksburg, VA) circulated dechlorinated water through the tank. Three times per week, ARC staff fed the frogs with adult *Xenopus* diet (Zeigler Bros. Inc., Gardners, PA), sprinkling the food (10 pellets/frog) into the tank and, after a few hours when the *Xenopus* had fed, removing excess food with a net. As judged necessary, the ARC staff partially changed out the water in the tank. Every 90 days, they moved the frogs into a newly cleaned tank that contained, in equal amounts, water from the previous tank and new de chlorinated water.

We anaesthetised frogs by immersion in a solution containing 0.2% tricaine (i.e., MS-222; ethyl 3-aminobenzoate methanesulfonate, catalogue # A5040, Sigma-Aldrich, St Louis, MO, USA). When an animal became unresponsive to touch, we removed it from the solution and surgically extracted the ovaries. The animal was euthanized by cardiac excision prior to recovery from anaesthesia.

In some experiments, we isolated oocytes from *Xenopus* ovarian lobes shipped overnight from NASCO.

**Rabbits.** We purchased adult female New Zealand white rabbits—named 'Ren' and 'Stimpy' for internal identification of the source of antiserum—from Charles River Laboratories (Ashland, OH). The rabbits were housed individually in stainless-steel cages in a temperature-controlled room (20–22°C) with a 12 h light/12 h dark cycle with free access to commercially available pelleted rabbit chow and fresh water. They were observed daily by trained animal care personnel in the Animal Research Centre (ARC) to monitor general health and well-being. Following IACUC guidelines, ARC staff obtained a 5-mL pre-immune blood sample from an ear vein and then inoculated two rabbits—gently restrained in a rabbit-restraint cage and lightly sedated

with intramuscular acepromazine (2 mg/kg)—with fresh DIDS-KLH fusion protein (see below[1]) and Freund's complete adjuvant via subcutaneous injection on the back of the rabbit. A 10-mL sample of blood (via ear veins, alternating sides) was taken 14–21 days after the first boost, and the sera were evaluated for antigenicity towards proteins labelled with DIDS. Once a rabbit began producing sufficient titers of polyclonal anti-DIDS antibodies, staff collected 10 mL of blood per kg of body weight (∼30–50 mL total) at intervals of 21 to 30 days. After two such larger blood collections, ARC staff euthanized the animals by cardiac excision and exsanguination, with blood from the terminal collection being transferred to our laboratory.

## Solutions and chemicals

**OR3 media (for maintaining oocytes).** We follow the protocol described in (Musa-Aziz et al., 2010). Briefly, we added to 1.8 L of $H_2O$ one pack of powdered Leibovitz L-15 media with L-glutamine (catalogue # L4386, Sigma–Aldrich), 100 mL of 10,000 U/mL penicillin/streptomycin solution (cat # 15 140 122, Thermo Fisher Scientific, Waltham, MA, USA; hereafter abbreviated TFS), and 5 mM HEPES free acid. We then titrated the solution to pH 7.50 with NaOH, periodically measuring an osmolality of 195 mOsm, sterile-filtered the solution using a Corning Disposable Vacuum Filter/Storage system (catalogue # 09-761-107, TFS), and stored it in a cold room for up to 2 weeks before use. Finally, just before use, we sterile-filtered a small volume of solution using a 60-mL syringe (catalogue # 14-955-461, TFS) and a sterile in-line Nalgene filter with a 0.22-μm pore size and a 25-mm diameter (catalogue # 723-9920, TFS).

**Solutions for physiology experiments.** Table 1 describes the final composition of nine solutions used in the physiology experiments. All were assembled and used at room temperature (∼22°C). Where required, we describe any necessary additional steps for correct solution assembly (e.g., the order in which components should be added) below.

**(Solution 3) 5% $CO_2$/33 mM $HCO_3^-$ for measuring $\Delta pH_S$ due $CO_2$ influx.** After adding all components except $NaHCO_3$, we titrated the solution to pH 7.50, then added 33 mM $NaHCO_3$, and then equilibrated the solution with 5% $CO_2$ (balanced with air), which brought pH back to 7.50.

**(Solutions 4 and 5) 0.5 mM $NH_3$/$NH_4^+$ in ND96 for measuring $\Delta pH_S$ due $NH_3$ influx.** We first made the 5 mM $NH_3$/$NH_4^+$ in ND96 solution (Solution #4) in which we replaced 5.0 mM NaCl with an equivalent amount of $NH_4Cl$, and then diluted this solution 1:10 into ND96 to create the final 0.5 mM $NH_3$/$NH_4^+$ solution (Solution #5).

**(Solution 6) Hypotonic ND96 for measuring $P_f$.** This is a variant of ND96 in which we reduced [NaCl] to lower osmolality to 80 mOsm/kg.

**(Solution 7) ND96+pCMBS.** In some experiments, we pre-incubated oocytes for 30 min in ND96 containing 1 mM pCMBS (catalogue # C367750, Toronto Research Chemicals, North York, Ontario, Canada), a sulfhydryl reagent, added to the solution as dry powder to ND96 (Solution #1).

**(Solution 8) DIDS.** In some experiments, we pre-incubated oocytes for 1 h in an ND96 containing 100 μM DIDS (catalogue # D3514, Sigma–Aldrich), an amino-reactive agent added to the solution as a dry powder to ND96 (Solution #1).

**(Solution 9) BSA (Bovine serum albumin, to scavenge DIDS).** In some experiments, after DIDS pretreatment, we washed off unreacted DIDS by exposing oocytes to an ND96 to which we added 0.2% BSA (catalogue # A9418, Sigma–Aldrich) directly to ND96 (Solution #1).

When assembling all solutions, we measured pH using a Ross electrode (catalogue # 927007MD, TFS), connected to a Dual Star pH meter (catalogue # 8102BNUWP, TFS) and calibrated with two standards (from TFS), buffer solution standards pH 6.0 (catalogue # SB104-1) and pH 8.0 (catalogue # SB112-20). We measured osmolality with a vapor-pressure osmometer (catalogue # 5520 Vapro, Wescor Inc., Logan, UT).

## Oocyte isolation

We placed ovarian lobes in a sterile 100-mm Petri dish containing ND96, cut them into irregular pieces (<1 cm in size) containing ∼10 oocytes each using small iridectomy scissors. We then poured the mixture into a new Petri dish containing ∼15 mL of $Ca^{2+}$-free ND96 and gently agitated the dish on a horizontal shaker[2] for 10 min, poured off as much liquid as possible, added fresh $Ca^{2+}$-free ND96, and repeated the pour/shake/wash cycle twice more. We then poured off the $Ca^{2+}$-free ND96 before adding ∼15 mL of

---

[1]See Methods > Generation of the DIDS antibody

[2]In some later experiments, we instead put the oocytes and $Ca^{2+}$-free solution into 50-mL Falcon tubes (catalogue # 14-959-49A, TFS) and agitated them using a Tube Rotator (Scientific Equipment Products, Baltimore, MD).

**Table 1. Solutions***

| Solution | 1 | 2 | 3 | 4 | 5 | 6 | 7 | 8 | 9 |
|---|---|---|---|---|---|---|---|---|---|
| Component or parameter | ND96 | Ca$^{2+}$-free ND96 | 5% CO$_2$/ 33 mM HCO$_3^-$ [for (ΔpH$_S$)$_{CO2}$] | 5 mM NH$_3$/NH$_4^+$ in ND96 | 0.5 mM NH$_3$/NH$_4^+$ in ND96 [for (ΔpH$_S$)$_{NH3}$] | Hypotonic ND96 [for $P_f$] | ND96+ pCMBS | ND96+ DIDS | BSA [to scavenge DIDS] |
| NaCl (mM) | 96 | 98.7 | 63 | 91 | 91 | 33 | 96 | 96 | 96 |
| KCl (mM) | 2.0 | 2.0 | 2.0 | 2.0 | 2.0 | 2.0 | 2.0 | 2.0 | 2.0 |
| CaCl$_2$ (mM) | 1.8 | 0 | 1.8 | 1.8 | 1.8 | 1.8 | 1.8 | 1.8 | 1.8 |
| MgCl$_2$ (mM) | 1.0 | 1.0 | 1.0 | 1.0 | 1.0 | 1.0 | 1.0 | 1.0 | 1.0 |
| HEPES (mM) | 5 | 5 | 5 | 5 | 5 | 5 | 5 | 5 | 5 |
| NaHCO$_3$ (mM) | 0 | 0 | 33 | 0 | 0 | 0 | 0 | 0 | 0 |
| NH$_4$Cl (mM) | 0 | 0 | 0 | 5 | 0.5 | 0 | 0 | 0 | 0 |
| CO$_2$ (%) | 0 | 0 | 5 | 0 | 0 | 0 | 0 | 0 | 0 |
| pH (adjusted with NaOH) | ~7.50 | ~7.50 | ~7.50 | ~7.50 | ~7.50 | ~7.50 | ~7.50 | ~7.50 | ~7.50 |
| Temperature (°C) | RT | RT | RT | RT | RT | RT | RT | RT | RT |
| Osmolality (mOsm) | ~195 | ~195 | ~195 | ~195 | ~195 | ~80 | ~195 | ~195 | ~195 |
| pCMBS (mM) | 0 | 0 | 0 | 0 | 0 | 0 | 1 | 0 | 0 |
| DIDS (µM) | 0 | 0 | 0 | 0 | 0 | 0 | 0 | 100 | 0 |
| BSA % | 0 | 0 | 0 | 0 | 0 | 0 | 0 | 0 | 0.2% |

*This table shows the compositions of all the solutions used in the physiology experiments the present paper. Abbreviations: BSA, Bovine serum albumin; DIDS, 4,4′-diisothiocyanatostilbene-2,2′-disulfonate; pCMBS, p-chloromercuribenzene sulfonate; $P_f$, osmotic water permeability.

a freshly-made mixture of $Ca^{2+}$-free ND96 and 2 mg/mL Collagenase Type IA (catalogue # **C9722**, Sigma–Aldrich), and gently agitating the dish on a horizontal shaker for 40 min. We next poured off the collagenase solution, added collagenase-free $Ca^{2+}$-free ND96, gently agitated the dish for 15 min, and repeated the pour/shake/wash cycle twice more. We then poured off the $Ca^{2+}$-free ND96, added ND96 (i.e., containing $Ca^{2+}$), gently agitated the dish on a horizontal shaker for 15 min, and repeated the pour/shake/wash cycle twice more. Finally, we poured off the ND96, added ~15 mL OR3 media, repeated the pour/wash cycle twice more, poured off most of the liquid, transferred the oocytes and remnant OR3 media to a fresh Petri dish containing fresh OR3 media, and incubated the oocytes in an incubator at 18°C. Later on the same day, we used a stereomicroscope to sort the oocytes and select individual, defolliculated, mature oocytes (stage V/VI), which we immediately moved to a 6-well plate containing OR3 media (as many as ~50 oocytes/well), using a fire-sterilised transfer pipette prepared by using a diamond pencil to score the tapered end of a Pasteur pipette at a diameter large enough to accommodate an oocyte, breaking off and discarding the thin end of the pipette, fire-polishing the cut end of the pipette, and attaching the large end of the pipette to a manual 2-mL Pipette Pump (catalogue # S3-594-3, TFS). We incubated the oocytes overnight at 18°C before injecting them the following day with *cRNA* or $H_2O$ (see below).

### cRNA synthesis

The cDNAs encoding both hAQP1-WT (accession# NM_198 098) and hAQP1-C189S were gifts of Dr. Peter Agre (Johns Hopkins University). We subcloned the open reading frames of the constructs into the expression plasmid pGH19 (Trudeau et al., 1995), a vector containing the 3′ and 5′ untranslated regions of the *Xenopus* laevis $\beta$-globin gene. In experiments in which we expressed hAQP1-WT or hAQP1-C189S in *P. pastoris*—as well as in some control experiments in oocytes—we added an N-terminal (Nt) FLAG-tag (MASEFKKKL…; FLAG sequence, underscored; hAQP1 sequence, double underscored).

We linearised cDNA constructs using NotI (for untagged hAQP1-WT and hAQP1-C189S) or XhoI (for the hAQP1-FLAG tagged versions). The linearised DNA was then purified using the QIAquick PCR purification kit (catalogue # 28 104, Qiagen Inc., Valencia, CA). We then synthesised capped RNA (cRNA) using T3 (for untagged constructs; catalogue # AM1348, Ambion, Austin, TX, USA ) or T7 (for FLAG-tagged constructs; catalog # AM1344, Ambion) mMessage mMachine kits (Ambion, Austin, TX, USA). The cRNA was purified using the RNeasy MinElute RNA Cleanup Kit (catalogue # 74 204,

Qiagen). The cRNA concentration was determined based on ultraviolet absorbance at 260 nm, and quality was assessed according to the A260/280 ratio and gel electrophoresis.

### Injection of oocytes with cRNA or water

After isolation, defolliculation and sorting (Day 0), we injected stage V–VI oocytes with 25 nL of cRNA (1 ng/nL) encoding hAQP1-WT, its C189S mutant, or the FLAG-tagged versions, or ~25 nL of water as a control (Day 1). The injection apparatus consisted of a Nanoject II Variable Volume Automatic Injector (Drummond Scientific Company, Broomall, PA) connected to an injection needle that we pulled from 10-µL micro-dispenser capillary glass tubing (catalogue # 3-000-210-G, Drummond), using a Model P-97 Flaming/Brown Micropipette Puller (Sutter Instrument Company, Novato, CA) and then modified by manually breaking the sealed end to yield a tip diameter of ~50 µm. After injection, we transferred oocytes to a 6-well plate (up to ~50/well) and maintained them in OR3 media and maintained them for 3 days in an incubator at 18°C before use in experiments (generally on Day 4).

### Electrophysiological recordings

For a schematic representation of the experimental chamber, position of the oocyte and arrangement of electrodes, see Fig. 8*B* in Musa-Aziz et al. (2010).

**Chamber and solution delivery.** We placed oocytes into the channel (3 mm wide × 2 mm deep × 30 mm long) of a plastic perfusion chamber. We delivered the physiological solutions—or pH standards for calibrating pH microelectrodes at the outset of an experiment (see next section)—at a constant, total flow of 4 mL/min at room temperature (~22°C), using a series of dual-syringe pumps (Model 22, Harvard Apparatus, South Natick, MA), each of which drove two 140-mL plastic syringes (Sherwood Medical, St. Louis, MO) of identical contents to deliver two solutions at 2 mL/min each.[3] The two solutions from each pump flowed through separate lengths of Tygon tubing (5/32-inch (~4.0 mm) outer diameter, 3/32-inch (~2.4 mm) inner diameter; Ryan Herco Products Corp., Burbank, CA; Formulation R3603-3; OD 4.8 mm/ID 1.6 mm to two parallel assemblies of pneumatically operated 5-way valves (Clippard Instrument Laboratory, Cincinnati, OH), daisy-chained in such a way that we could switch amongst

---

[3]The apparatus had this dual-solution arrangement to enable the use of out-of-equilibrium $CO_2/HCO_3^-$ solutions in experiments that are not part of the present study.

identical pairs of solution lines. The output of the parallel switching assemblies flowed into two lengths of Tygon tubing that carried the solutions to the vicinity of the chamber, where the two lines merged in a 'mixing T', the output of which flowed into one end of the chamber channel. A suction device removed the solution at the opposite end of the channel. See Musa-Aziz et al. (2014a, b) for details.

**Measurement of membrane potential and intracellular pH.** As described previously (Musa-Aziz et al., 2010, 2014a), we impaled the oocyte (with the dark animal pole facing upward) with two microelectrodes, one for measuring membrane potential and the other for measuring intracellular pH. Both microelectrodes, with tip diameters of ∼1 μm, we pulled from thin-walled borosilicate microfilament glass tubing (catalogue # G200TF-4, 2.0 mm OD ×1.56 mm ID, Warner Instruments Corporation, Hamden, CT), using the aforementioned P-97 microelectrode puller.

The microelectrodes for measuring membrane potential ($V_m$) were filled with 3M KCl. We inserted an electrode into a half-cell/microelectrode holder (model # ESW-F20N, Warner Instruments Corp.), attached the holder to the $V_m$ probe of an OC-725 two-electrode Oocyte Clamp (Warner Instruments Corp.), and mounted the probe on a model MM-33R micromanipulator (Warner Instruments Corp.). The $V_m$ microelectrodes had resistances of ∼0.3–∼0.6 MΩ.

The microelectrodes for measuring intracellular pH (pH$_i$) were identical to the $V_m$ electrodes except for the following fabrication process. In order to remove moisture, we placed the pulled micropipettes, with tips upwards, into the cylindrical holes of a reusable aluminum block (machined to create 'legs' and thereby allow gases to circulate from beneath, and with smaller concentric cylindrical holes extending the entire height of the block to allow access of gases from below, to the interior of the glass capillary), placed the block into the bottom of a 100-mm Pyrex petri dish, covered the electrodes and block with an inverted 250-mL beaker, placed the petri dish on an open metal rack of an oven at 200 °C, and then baked overnight. With the microelectrodes still in the 200 °C oven, we then silanized them by tilting the beaker slightly and depositing 80 μL of *bis*-di-(methylamino)-dimethylsilane (Sigma–Aldrich, catalogue #14 755) between the 'legs' of the aluminum block, releasing the beaker, and allowing the silane fumes to interact with the microelectrodes for 40 min before removing the beaker. The silanized electrodes were cured in the oven at 200 °C until ready for use. We used a hand-drawn, soft-glass pipette with a long tip to backfill each pH$_i$ electrode with a liquid pH-sensitive sensor (catalogue # 95 293, Hydrogen Ionophore I, mixture B, Fluka Chemical Corp., Ronkonkoma, NY), as

described by Ammann et al., 1981, creating a column that extended ∼1 mm from the microelectrode tip. We then backfilled a microelectrode with a buffer solution (containing, in mM, 40 $KH_2PO_4$, 23 NaOH, 15 NaCl, adjusted to pH 7.0), inserted the electrode into a half-cell/microelectrode holder (model # ESW-F20N, Warner Instruments Corporation), attached the holder to one probe of a model FD223 dual high-impedance electrometer, World Precision Instruments (WPI), Inc., Sarasota, FL), and mounted the probe on a model MM-33R micromanipulator (Warner Instruments Corp.). The pH$_i$ microelectrodes had resistances of ∼0.3–0.6 MΩ.

After placing the tips of the $V_m$ and pH$_i$ microelectrodes in ND96 solution in the chamber channel (which we now refer to as the 'bath'), we obtained $V_m$ by an analogue electronic subtraction of the system ground of the OC-725 amplifier from the signal of the $V_m$ electrode. We similarly obtained the voltage due to pH$_i$ by electronically subtracting the signal of the $V_m$ electrode from the signal of the pH$_i$ electrode. For a schematic representation of the configuration of the electronics for obtaining $V_m$ (this section) and the voltages due to pH$_i$ (this section) and pH$_S$ (next section), see Fig. 7*C* in Musa-Aziz et al. (2010). The device that performed the subtractions (Yale University Subtraction Amplifier, v3.1) also appropriately scaled the voltages for the inputs of an analogue-to-digital converter, installed in a Windows-based computer. With the tips of the $V_m$ and pH$_i$ electrodes (and also the pH$_S$ electrode, simultaneously; see next section) in the bath, we obtained the slope of the pH$_i$ electrodes as we used the aforementioned solution-delivery system (see previous section) to flow a pH standard at pH 6.0 (catalogue # SB104-1) continuously through the chamber channel, recorded the voltage due to that pH, used the valve system to switch the flowing solution to a second pH-8.0 standard (catalogue # SB112-20), and then recorded the corresponding voltage. We typically switched between standards in a pH 6 → 8 → 6 → 8 sequence, accepting pH$_i$ microelectrodes that completed their electrical response—which also included the time for the solution in the chamber channel to fully change composition—within ∼5 s (∼2 s for pH$_S$ electrodes) and had slopes in the range of 53 to 60 mV/pH unit. After the recordings in the final pH-calibration solution, we reintroduced the ND96 solution into the bath and then added the oocyte to the chamber channel. After impaling the oocyte with the $V_m$ electrode, and with the tip of the pH$_i$ electrode near the oocyte, we obtained a single-point pH calibration of the pH$_i$ microelectrode in the ND96 solution (defined as having a pH of 7.50). We then impaled the oocyte with the pH$_i$ microelectrode. After stabilisation, oocytes had spontaneous $V_m$ values at least as negative as –40 mV. We continuously recorded pH$_i$ to judge the integrity of the oocyte and compare the pH$_i$ time course with that of surface pH. In the present paper,

we are not presenting the pH$_i$ data, which nevertheless are similar to those of our previous studies (Musa-Aziz, Chen et al., 2009; Musa-Aziz et al., 2010; 2014a, 2014b).

**Measurement of surface pH.** We measured surface pH with liquid-sensor pH microelectrodes that we fabricated and employed in the same way as we did for pH$_i$ microelectrodes (see above), but with four differences. First, we pulled the pH$_S$ microelectrodes from standard-walled (rather than thin-walled) borosilicate microfilament glass tubing (Part No. G200F-4, 2.0 mm OD ×1.16 mm ID, Warner Instruments). Second, we used a microforge to break off and fire-polish the tips (inner diameter $\cong$ 20 μm at tip) as one would for a giant-patch pipette (Musa-Aziz et al., 2010). It was to facilitate the fire polishing that we used the standard-walled glass tubing. Third, we used a vented microelectrode holder (model # ESW-F20N, Warner Instruments Corp.) mounted on a model MM-33L micromanipulator. The vent prevented pressure buildup that would otherwise have pushed the liquid pH sensor out of the wide electrode tip as we pushed the pH$_S$ electrode into the holder. Note that we attached the microelectrode holder of the pH$_S$ electrode to the second of two inputs of the FD223 electrometer. And fourth, after obtaining the pH$_S$-electrode slope (which we did at the same time as obtaining the pH$_i$ electrode slope, as described above) and single-point calibration of the pH$_S$ electrode in ND96, we used an ultra-fine computer-controlled micromanipulator (model MPC-200 system, Sutter Instrument Company) to position the blunt tip of the pH$_S$ electrode near the oocyte's equator (i.e., between the upward-facing animal pole and the downward-facing vegetal pole), and ~5 degrees behind the meridian (i.e., barely in the 'shadow' of the flowing extracellular solution) until the pH$_S$ electrode tip just touched the surface of the oocyte (Musa-Aziz et al., 2010, 2014a). We then further advanced the tip by ~40 μm to create a slight dimple in the membrane. Periodically during the experiment (indicated by vertical grey bands in each panel of Figs 2*B*, 3 and 6), we withdrew the electrode ~300 μm from the oocyte for recalibration in the bulk extracellular fluid (bECF) of the bath (i.e., pH 7.50).

The external reference electrode for the pH$_S$ measurement was a calomel half-cell, bridged to the chamber channel via a long glass micropipette filled with 3M KCl, connected to a model 750 electrometer (WPI), and positioned so that its tip was just downstream of the oocyte. We obtained the pH$_S$ signal by an analogue electronic subtraction of the calomel signal from the signal of the pH$_S$ electrode.

We established a virtual ground with an Ag/AgCl half cell (connected to the $I_{Sense}$ input of the voltage-clamp amplifier) bridged to the chamber by a second glass micro-electrode filled with 3M KCl, and positioned the tip of this second bridging pipette close to that of the first, that is, just downstream of the oocyte.

**Measurement of osmotic water permeability.** We measured $P_f$ as described by Preston et al. (1992, 1993) and Zhang et al. (1990). Using an approach similar to those in previous studies by our group (Musa-Aziz, Chen et al., 2009; Virkki et al., 2001), we dropped an oocyte into a Petri dish containing a hypotonic solution (80 mOsm; Solution #6, Table 1) to induce cell swelling and, whilst illuminating from beneath the dish, used a video camera to obtain images of the oocyte (1/s × 60 s). Using a small metal sphere next to the oocyte as a size reference, we determined the projection areas of the oocyte, and—assuming the oocyte to be a perfect sphere—computed idealised oocyte volume ($V_{Oocyte}$) and idealised oocyte surface area as a function of time. We estimated the actual surface area ($S$, in cm$^2$) by multiplying the idealised surface area by a factor of 9 (Chandy et al., 1997). We computed $P_f$ from the following equation:

$$P_f = \underbrace{\frac{\left(\frac{dV_{Oocyte}}{dt}\right)_{max}}{S \times \Delta Osm \times V_W}}_{\text{units: } cm/s},$$

the form of which differs slightly from that of Zhang et al. (1990).[4] Here, $(dV_{Oocyte}/dt)_{max}$ is the maximum time rate of change of cell volume (i.e., the maximal H$_2$O influx, or $J_{V,max}$, in cm$^3$ s$^{-1}$), $\Delta Osm$ is the initial osmotic gradient across the oocyte membrane (i.e., 195−80=115 mOsm, expressed as mol cm$^{-3}$), and $V_w$ is the molar volume of water (18.1 cm$^3$ mol$^{-1}$). As for the electrophysiological studies, we performed all $P_f$ assays at room temperature (~22°C).

## Protocols for physiological experiments

Fig. 1*A*–*J* illustrates the sequence of events in our experimental protocols.

---

[4]To obtain $P_f$ in the proper units (cm s$^{-1}$) in the above equation, the derivation by Zhang et al. (1990) requires that the division of 'moles of osmotically active particles' (in the $\Delta Osm$ term) by 'moles of water' (in the $V_W$ term) yield a numerical result that is unitless, which of course is impossible. This inconsistency arises during the derivation of the water-flux equation when one switches from describing the energy of water (i.e., RT) as 'per mole of water' per se to the energy of water 'per mole of osmotically active solute' during the introduction of the Van't Hoff equation. The introduction of a phantom conversion factor with a value of unity—but with the units '(per mole of osmotically active solute)/(per mole of water)'—eliminates the inconsistency.

**Assessing oocyte health.** Because hAQP1, with its encoding cRNA injected as described above, tends to stress the oocytes, we generally performed experiments on Day 4. We judged the health of the oocytes by microscopic observation of shape and colour, by $V_m$ (more negative than –40 mV, as described above), and by a firm membrane as we pushed the $pH_S$ electrode up against the surface (Wang et al., 2025).

**Electrophysiological assays.** On the day of an experiment, we followed a 10-step protocol:

(A) Use a transfer pipette to pick up an untreated oocyte from a well of a 6-well plate (each well containing OR3 media + ∼30 oocytes, all injected with the same material) and drop the oocyte into a Petri dish containing ND96 (Solution #1, Table 1). After flushing the transfer pipette tip several times with ND96 (to remove OR3).

(B) Use the transfer pipette to move the washed oocyte to an identified well of a 24-well plate (1 oocyte/well) containing ND96. Repeat this procedure for several oocytes, each of which will remain in its well (1–3 h) until we proceed to the electrophysiological assay.

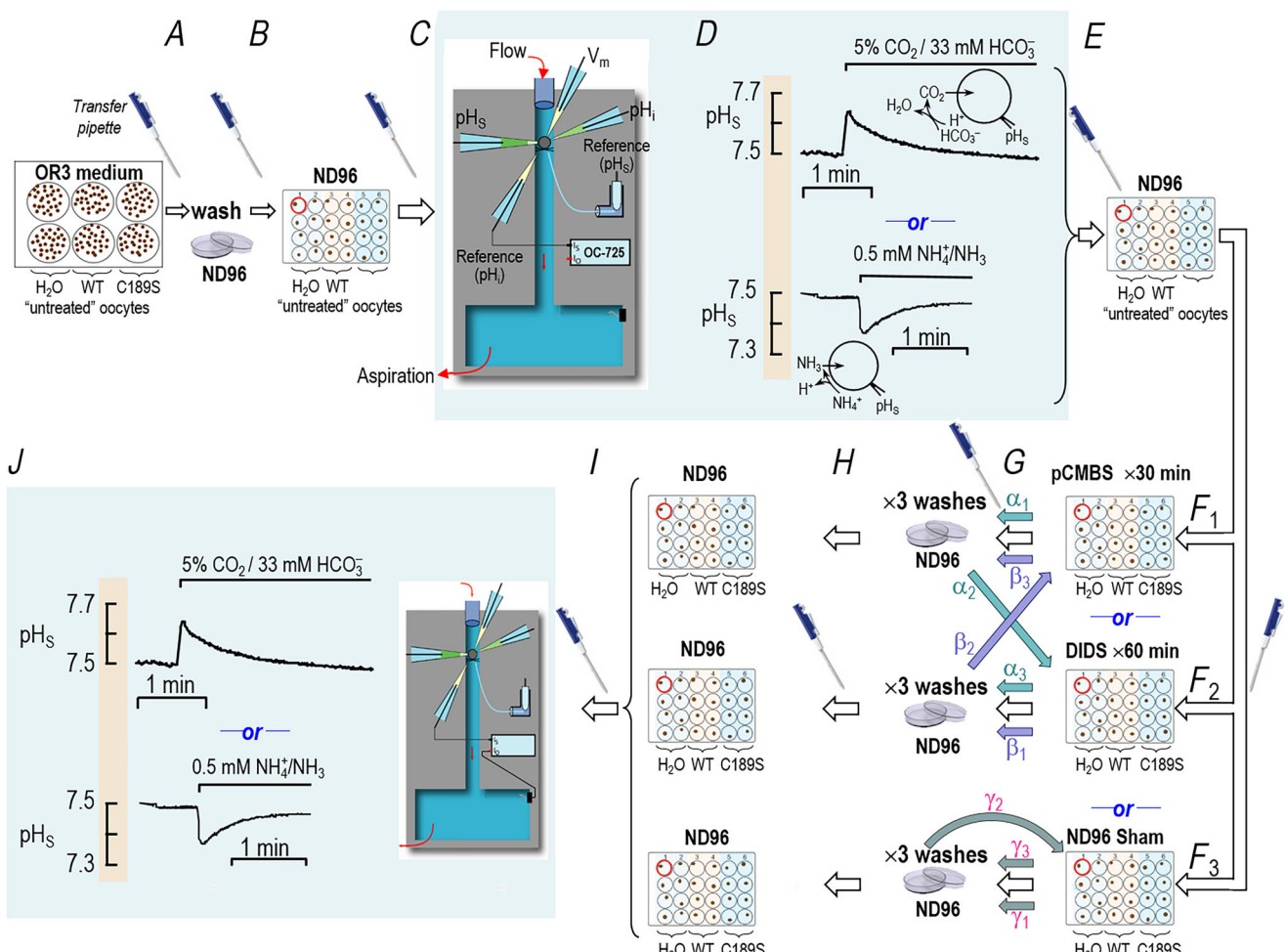

**Figure 1. Flow chart showing experimental protocols**
The letters 'A', 'B', 'C' and so on are above broad arrows and near icons of a pipettor that identify the transfer of an oocyte (using a transfer pipette) from one step in our protocol to another. 'D' and 'J' are the initial and final electrophysiological recordings, respectively. The oocytes may be controls (injected with H₂O) or those expressing either hAQP1-WT (AQP1) or hAQP1-C189S (C189S). In transfer 'F', we move the oocyte to a solution containing pCMBS (F₁), DIDS (F₂), or no drug (F₃). As indicated by the white arrows, we can directly transfer from F₁ (or F₂ or F₃) to G to H. Alternatively, at transfer 'F', we can execute subsidiary protocols (α/teal arrows, β/lavender arrows, γ/grey arrows) in which we return oocytes from washes in ND96 back to a solution containing an inhibitor (i.e., pCMBS or DIDS) or ND96 (i.e., sham drug exposure). Thus, an oocyte might undergo the teal-colored pCMBS/DIDS sequence (α₁→α₂→α₃), the lavender-colored DIDS/pCMBS sequence (β₁→β₂→β₃), or the grey-colored sham/sham sequence (γ₁→γ₂→γ₃). After the transfer H, we perform the final electrophysiology recordings.

(C) Use the transfer pipette to place the oocyte in the chamber, begin flowing ND96, and measure $V_m$, $pH_i$, and $pH_S$ under basal conditions.

(D) Monitor $pH_S$ as we replace ND96 with either the 5% $CO_2$/33 mM $HCO_3^-$ solution (Solution #3, Table 1; as in Figs 2*B* and 3) or the 0.5 mM

$NH_3$/$NH_4^+$ solution (Solution #5, Table 1; as in Figs 5*B* and 6); all oocytes in the study underwent this control assay.

(E) Immediately upon completion of the $pH_S$ assay, use a transfer pipette to return the oocyte from the chamber to its original well and incubate the oocyte at

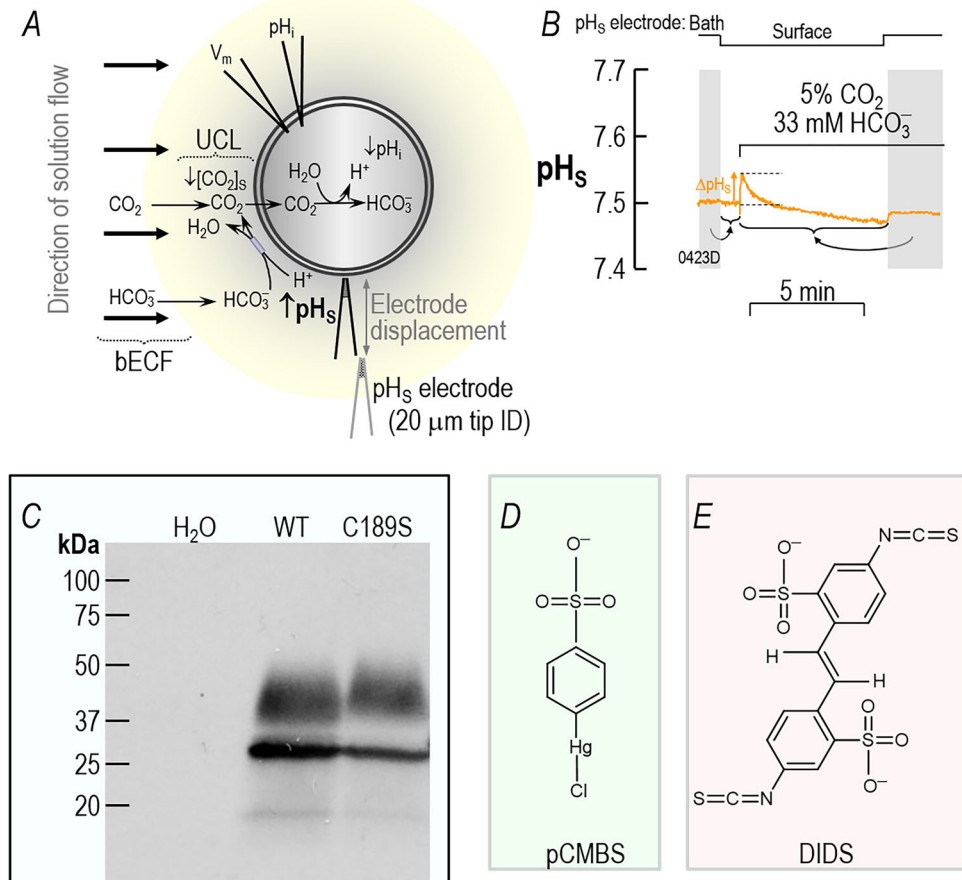

**Figure 2. $CO_2$ influx into an oocyte.**
A, schematic illustration of a $CO_2$-influx experiment. Thick black arrows indicate the direction of convective flow within the bulk extracellular fluid (bECF). Thinner arrows indicate solute diffusion or reactions. The maize-colored halo indicates the layer of extracellular unconvected fluid (EUF). At the outer surface of the membrane, $CO_2$ influx creates a $CO_2$ deficit, in part replenished by the reaction $HCO_3^- + H^+ \rightarrow CO_2 + H_2O$ (which raises $pH_S$) and in part replenished by diffusion from the bECF (an isohydric process). The double-headed arrow indicates displacement of the $pH_S$ electrode from the cell surface to the bECF for recalibration. ID, inner diameter; UCL, unconvected layer. B, sample surface-pH ($pH_S$ record). The upward arrow indicates the maximal change in $pH_S$ ($\Delta pH_S$) during the application of $CO_2$/$HCO_3^-$. During the two periods indicated by the vertical grey bands, the tip of the $pH_S$ electrode was displaced ~300 μm away from the cell surface for recalibration in the bulk extracellular fluid (i.e., pH = 7.50). The curved grey arrow points to horizontal braces that indicate the portions of the experiment to which each calibration pertains. The filename for this representative trace, '0423D'—that we reuse in Fig. 3*A*—is annotated at the bottomleft corner of the panel. C, western blot, probed with an anti-AQP1 antibody, showing relative expression of hAQP1-WT versus hAQP1-C189S in one of 3 membrane preparations of oocytes, each from a different donor frog and different cRNA injections. For each blot, we used the ImageJ gel-analyzer plug-in to group bands at all molecular weights in each lane to quantify aggregate band intensities. For each blot, we normalised the hAQP1-WT intensity to 100%, and expressed hAQP1-C189S intensity as a fraction of hAQP1-WT. The analysis showed that hAQP1-C189S expression in the 3 blots was 72.5%, 59.8% and 117.7% of WT; thus, the average was 83.3%±30.4% of WT (mean±SD). A Student's *t*-test with Welch's correction yielded a *P*-value of 0.44, indicating no significant difference between hAQP1-WT versus hAQP1-C189S mutant expression levels. D, structural formula of pCMBS. E, structural formula of DIDS.

∼22°C in ND96. We serially process all oocytes from a 24-well plate up to this stage (i.e., steps 'C' through 'E'), and then pause. Thus, this second period of incubation in ND96 (step 'E') ranges from ∼30 min to ∼3 h.

(F) Use a transfer pipette to move each oocyte in the 24-well ND96 plate to a corresponding well in one of three new 24-well plates. Here, the wells contain one of the following three solutions: ($F_1$) 1 mM pCMBS dissolved in ND96 (Solution #7, Table 1) for 30 min incubation; see Figs 3*D–F*, 6*D–F* and 7–9*C, D*); ($F_2$) 100 µM DIDS dissolved in ND96 (Solution #8, Table 1; 60 min incubation; see Figs 3*G–I* or 6*G–I*); or ($F_3$) ND96 without an added drug ('sham').

(G) Wash the oocyte. Except in a few cases (see indented paragraph below), we wash the oocyte ×3 in a petri dish containing ordinary ND96, as follows. We use a transfer pipette to pick up an oocyte from a 24-plate (step 'F') and release the oocyte into a petri dish containing ND96, quickly pick up and release the oocyte into this wash solution twice more (to remove excess drug, if present). At this point, most oocytes progress immediately through step 'H' (below). However, in a minority of experiments (see Fig. 3*J–L*), we serially treated oocytes with both pCMBS and DIDS. Here, oocytes previously treated with pCMBS (step '$F_1$') we exposed to DIDS, as indicated by steps $\alpha_1 \rightarrow \alpha_2 \rightarrow \alpha_3$. Conversely, oocytes previously treated with DIDS (step '$F_2$') we exposed

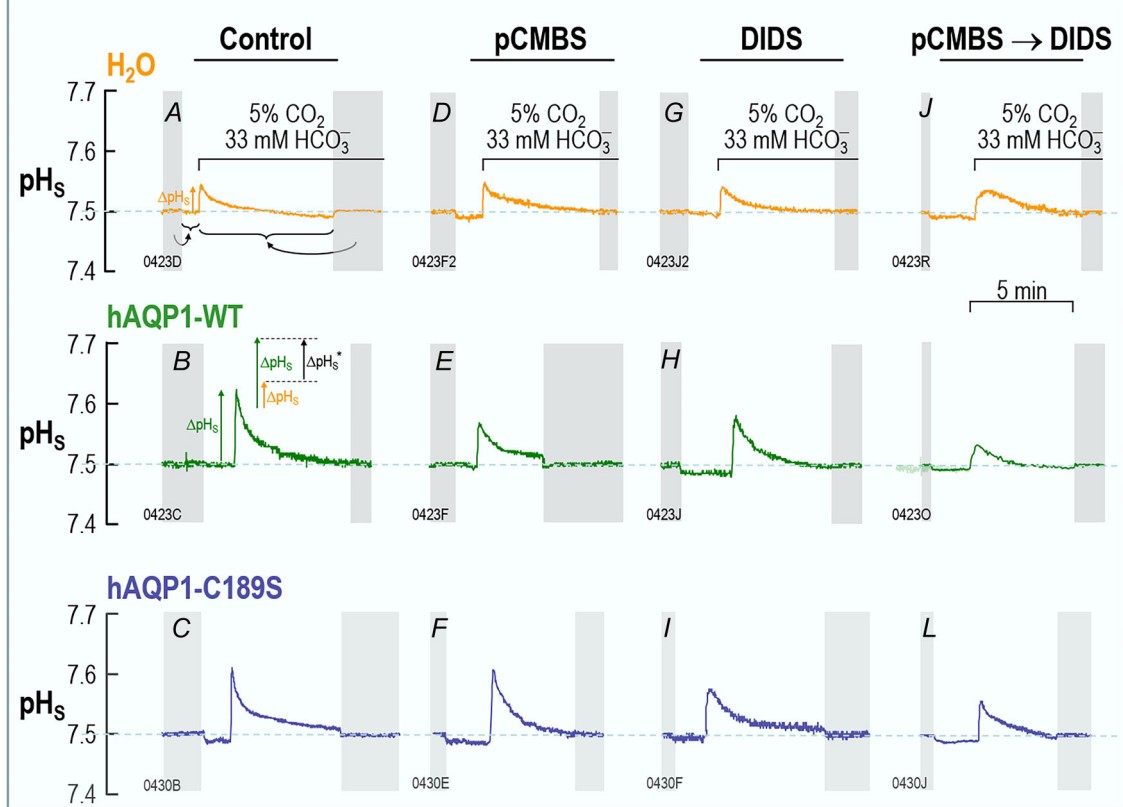

**Figure 3. Surface-pH transients triggered by a CO₂/HCO₃⁻ exposure and subsequent CO₂ influx**
The figure shows representative pH$_S$ records from oocytes injected with H₂O (top row) or cRNA encoding hAQP1-WT (middle row) or its C189S mutant (bottom row), and later exposed first to the ND96 solution and then, at the indicated times, to a solution containing CO₂/HCO₃⁻. All oocytes underwent the 'Control' protocol, followed by one or two of the three indicated $F_1$/$F_2$/$F_3$ protocols in Fig. 1, although some oocytes did not survive beyond the control protocol. As necessary, we pre-incubated oocytes in 1 mM pCMBS for 30 min or in 100 µM DIDS for 1 h. Neither drug was present in the bulk solution at the time of the assays. The two grey bars in each panel indicate when we withdrew the pH-electrode tip from the oocyte surface to the bulk extracellular fluid (pH 7.50) for recalibration. In panel A, the curved grey arrows point and horizontal braces have the same meanings as in Fig. 2*B*, and the upward gold arrow indicates the control ΔpH$_S$. In panel B, the upward green arrow indicates the ΔpH$_S$ in an oocyte expressing hAQP1. Statistics Table 3 presents the analysis of the mean ΔpH$_S$ magnitude recorded from all H₂O, hAQP1-WT or hAQP1-C189S 'control' oocytes for which we show representative traces in panels *A*, *B* and *C*. The inset shows that the difference in heights of the green arrow and the gold arrow (its day-matched control) is the channel-dependent ΔpH$_S$ (ΔpH$_S$*).

to pCMBS, as indicated by steps $\beta_1 \rightarrow \beta_2 \rightarrow \beta_3$. In one experiment without any drugs, we also followed the steps $\gamma_1 \rightarrow \gamma_2 \rightarrow \gamma_3$ ('double sham').

Not illustrated in Fig. 1 is a protocol variant that we employ in a few cases (i.e., Fig. 9*A* and *B*/right bars) in which we introduce an additional step between 'F$_2$' and 'G'. Here, after treating an oocyte with DIDS (step 'F$_2$'), we scavenge unreacted DIDS with albumin as follows. After using a transfer pipette to pick up the oocyte from the DIDS-containing ND96 solution, we release the oocyte into a Petri dish containing 0.2% albumin in otherwise DIDS-free ND96 (Solution #9, Table 1), and allow the oocyte to remain in the albumin solution for 20 min. We then wash off the albumin in step 'G' by transferring the oocyte to a Petri dish containing ordinary ND96, and pick up and release the oocyte ×3 using a transfer pipette.

(A) Use a transfer pipette to move the oocyte from the Petri dish (step 'G') to an identified well in one of three new 24-well plates, all containing ND96, in preparation for the next assay.

(B) Transfer the oocyte back to the chamber and (as in step 'C'), and again measure the new basal $V_m$, pH$_i$, and pH$_S$.

(C) Monitor (as in step 'D'), for the second time, the pH$_S$ transient as we replace ND96 with the CO$_2$/HCO$_3^-$ (or NH$_3$/NH$_4^+$) solution.

Thus, oocytes in the top row of Fig. 3 (i.e., Fig. 3*A*, *D*, *G* and *J*) went through one of the following two sequences: Panels A→D or A→J, which requires 12–18 h for an entire group of 24 oocytes in a day's work. Not all oocytes survived the entire protocol (as judged by the colour/integrity of the animal pole).

*P*$_f$ **assays.** On the same day that we performed our pH$_S$ assays, we also performed $P_f$ assays on separate cells from the same oocyte batch (i.e., oocytes prepared from the same ovary). We: (1) Use a transfer pipette to remove 6 untreated oocytes of each experimental group—hAQP1, C189S mutant, or hAQP1 FLAG-tagged—from an identified 6-well plate containing ND96 (i.e., a total of 18 oocytes). (2) Place the oocytes—as quickly as possible—in a Petri dish containing hypotonic ND96 (Solution #6, Table 1). (3) Use a video camera (described above) to monitor cell swelling at 1 image per second over a period of 60 s; oocytes of each group underwent this control assay. (4) Immediately remove the oocytes from the hypotonic solution to identified wells in another Petri dish containing standard ND96 solution to wash the oocytes ×3 at ~22°C (described above under electrophysiological assays). (5) Incubate the oocytes at ~22°C either for (5a) 30 min in ND96 containing 1 mM pCMBS (Solution #7, Table 1), or (5b) 1 h in ND96 containing 100 μM DIDS (Solution #8, Table 1). (6)

Wash the oocytes (as described under electrophysiological assays, point '5a'). (7) Transfer the oocytes to a Petri dish containing hypotonic, drug-free ND96 as in '2' above. (8) Monitor swelling as in '3' above.

### Generation of the DIDS antibody

**Generation of DIDS-fusion protein.** We used two rabbits to produce polyclonal antibodies against DIDS, as described by Garcia and Lodish (1989). Briefly, we reacted 2.4 mM DIDS (catalogue # D3514, Sigma–Aldrich), with 5 mg keyhole limpet hemocyanin (KLH; catalogue # H7017, Sigma–Aldrich) in 500 μL of phosphate-buffered saline (PBS; i.e., 150 mM NaCl, 10 mM phosphate buffer, pH 7.0) in the dark at 37°C × 40 min. Afterwards, we centrifuged the sample × 30 s in an Eppendorf benchtop centrifuge (catalogue # 5415C, TFS) to remove large insoluble aggregates. After pooling yellow supernatants from several centrifuge tubes, we dialysed against PBS at 4°C overnight in a Slide-A-Lyzer dialysis cassette (TFS). The following day, we removed the KLH-DIDS sample from the cassette; quantified the DIDS by absorbance spectroscopy at 340 nm, assuming a molar extinction coefficient ($\varepsilon$) of 54,000 M$^{-1}$ cm$^{-1}$; aliquoted the dialyzed KLH-DIDS solution; and immediately used it for immunisation.

**Processing of rabbit blood samples.** For both rabbits, we separately processed each of the three blood samples (see above[5])—2 lots of ~30–50 mL plus the terminal sample. We centrifuged the blood for 10 min at 1000 ×g using an Eppendorf bench top centrifuge to isolate the serum, and then evaluated antigenicity on western blots of DIDS-labelled hAQP1. We purified the antibodies (mainly IgG) from the remainder of the serum using a Protein A column (catalogue # 20,356, TFS), following the manufacturer's instructions; estimated [IgG] by measuring absorbance at 280 nm (assuming $\varepsilon = 210{,}000$ M$^{-1}$ cm$^{-1}$) using a NanoDrop 2000c spectrophotometer (TFS); separated the material into aliquots corresponding to 1 mg/mL of anti-DIDS antibody; and stored the aliquots at either –20°C for short-term storage or –80°C for long-term storage.

We evaluated the antigenicity using western blots of DIDS-labelled hAQP1-WT. For the purposes of this study, the antibodies used for detecting DIDS-labelled hAQP1-WT were from one lot (i.e., antibodies purified from the sera of one bleed).

---

[5]See Methods > Ethical approval and animal procedures > Rabbits

## Biochemical experiments

**Isolation of oocyte membranes.** We used a previously described method (Leduc-Nadeau et al., 2007), except that, after disrupting oocytes and centrifuging, we removed the supernatant and added fresh buffer to the pellet, repeating the procedure until the supernatant was clear (i.e., devoid of yolk).

**Purification of oocyte membrane proteins.** We resuspended isolated oocyte membranes in Tris-buffered Saline (TBS, 50 mM Tris HCl, pH 7.4, 150 mM NaCl) + 2% n-dodecyl-$\beta$-D-maltopyranoside (DDM; Sol-Grade, catalogue #D310S, Anatrace, Maumee, OH), incubated at 4°C × 2h, centrifuged at 16,000 × $g$ at 4°C × 30 min, and then collected the supernatant containing the solubilized membrane proteins and diluted it so that [DDM] was ≤ 0.5%.

**Overexpression of hAQP1-FLAG in Pichia pastoris.** We ligated N-terminally FLAG-tagged hAQP1 (<u>MDYKDDDDK</u><u><u>ASEFKKKL</u></u>; FLAG sequence underscored, hAQP1 sequence double underscored) into the pPICZ-A vector (Invitrogen, Carlsbad, CA), induced protein expression with methanol × 72 h at 26 °C, harvested the yeast by centrifugation, and obtained spheroplasts using zymolyase treatment (60 U/mL in 1 M sorbitol, 1 mM EDTA, 10 mM citrate buffer, pH 5.8) as previously described (Gustin et al., 1998). We then equally divided the spheroplasted cells into two groups (i.e., no DIDS and added 100 μM DIDS), incubated × 1 h at ~22 °C, and then solubilised overnight at 4°C in TBS + 2% DM, with cOmplete Protease Inhibitor Cocktail (catalogue #11 697 498 001, Roche, Mannheim, Germany). Following membrane solubilisation, we clarified the material by low-speed centrifugation, mixed with anti-FLAG resin (catalogue #A2220, Sigma–Aldrich), and eluted proteins from a column using TBS + 0.1% DM + 500 μg/mL FLAG peptide (catalogue #F4799, Sigma–Aldrich) at pH 7.4. In some cases, we subjected FLAG-purified hAQP1 to size-exclusion chromatography using a Sephadex G-75 column.

**Western blotting of membrane proteins from oocytes or Pichia pastoris.** We separated proteins extracted from oocyte membranes by SDS-PAGE using Tris-Glycine 4–20% gels (BioRad Laboratories, Hercules, CA) or proteins extracted from *Pichia* membranes using 12% Tris-Glycine gels; transferred the proteins to poly-vinylidene difluoride (PVDF) membranes using an iBlot 7-Minute Blotting System (TFS); and subsequently blocked the cross-linked proteins with Tris-buffered saline with Tween (TBST), comprising 25 mM Tris at pH 7.4, 150 mM NaCl, 0.05% Tween-20 (catalogue # P9416, Sigma–Aldrich)+5% milk powder × 1 h.

We probed blots from oocyte samples with a polyclonal anti-AQP1 (catalogue # AQP11-A, Alpha Diagnostics, San Antonio, USA), and blots from *P. pastoris* samples with monoclonal anti-FLAG (catalogue # F3165-2MG, Sigma–Aldrich) or polyclonal anti-DIDS (made in-house, see above), applying all primary antibodies at 1:1000 dilution from their stocks in TBST + 5% milk, at 4°C overnight. The next day, we washed the blot ×5 with TBST × 10 min. We detected the primary poly-clonal antibodies (i.e., anti-AQP1 and anti-DIDS) with an HRP-conjugated goat anti-rabbit secondary antibody (catalogue # 1 706 515, BioRad Laboratories), and detected the primary monoclonal antibody (i.e., anti-FLAG) with an HRP-conjugated goat anti-mouse secondary anti-body (catalogue # STAR207P, BioRad Laboratories). We applied both secondary antibodies at 1:5000 dilution from their stocks in TBST × 1 h, washed the blot ×5 with TBST × 10 min, developed the immunoblots using ECL Prime Western Blotting Detection Reagent (catalogue # 12 316 992, GE Healthcare Amersham, Piscataway, NJ), acquired images with a Typhoon Trio gel-documentation system (GE Healthcare), and evaluate band density patterns using Image J software (NIH, Bethesda, MD).

## Statistics

We present data as the mean±S.D. (standard deviation), and define N as the number of different frogs used and n as the number of replicate experiments (i.e., oocytes) for each condition in each dataset. We define $m$ as the number of comparisons in each dataset. Statistical analyses were performed using Origin 2024 software. Statistical comparisons amongst means were performed using a one-way analysis of variance (ANOVA) amongst all groups, followed by Holm–Bonferroni mean comparison (Holm, 1979) to control for type I errors across multiple comparisons, establishing the familywise error rate (FWER) at $\alpha = 0.05$. In brief, the unadjusted $P$-values for all $m$ comparisons in each dataset are listed from lowest to highest. In the first test, we compare the smallest unadjusted $P$-value to the first adjusted $\alpha$ value, $\alpha/m$. Upon rejection of the null hypothesis, we compare the second-smallest $P$-value to the second adjusted $\alpha$ value, $\alpha/(m-1)$, and continue the process accordingly. If the unadjusted $P$-value is ≥ the adjusted $\alpha$ at any stage, the null hypothesis is accepted, rendering all the subsequent hypotheses in the test group null. We conducted one-sample $t$-tests ($\alpha = 0.05$) to ascertain whether the channel-dependent $(\Delta pH_S{}^*)_{CO2}$ was significantly different from zero when oocytes were treated with both pCMBS and DIDS. If $P<0.05$, the

$(\Delta pH_S^*)_{CO2}$ was considered significantly greater than zero. We conducted one-sample *t*-tests ($\alpha = 0.05$) to ascertain whether the channel-dependent $(\Delta pH_S^*)_{NH3}$, was significantly different from zero when oocytes were treated with pCMBS. If $P<0.05$, the $(\Delta pH_S^*)_{CO2}$ was considered significantly greater than zero.

## Results

### Surface-pH measurements for hAQP1-WT and hAQP1-C189S

**pH$_S$ measurements for CO$_2$ transport.** Figure 2*A* is a schematic representation of the reaction and diffusion events that take place as we switch the bath solution—that is, the bulk extracellular fluid—from ND96 to another that contains 5% $CO_2$/33 mM $HCO_3^-$ at a constant pH of 7.50. This solution change not only leads to a net influx of $CO_2$ that lowers intracellular pH (Roos & Boron, 1981) but also lowers $CO_2$ concentration near the outer surface of the cell membrane ($[CO_2]_S$). This decrease in $[CO_2]_S$ (1) creates a $CO_2$ gradient from the bECF to the cell surface, which leads to partial $CO_2$ replenishment at the cell surface but does not alter pH; and (2) drives the reactions $HCO_3^- + H^+ \rightarrow H_2CO_3 \rightarrow CO_2 + H_2O$ at the cell surface, which also partially replenishes $CO_2$. These reactions cause an alkaline pH$_S$ transient, the maximal excursion of which we define as $\Delta pH_S$ (Endeward et al., 2006; Geyer et al., 2013; Musa-Aziz, Chen et al., 2009). Our colleagues have used 3D reaction-diffusion models to simulate these pH$_S$ and pH$_i$ changes as $CO_2$ diffuses into a spherical cell (Calvetti et al., 2020; Musa-Aziz et al., 2014a, 2014b; Occhipinti et al., 2014; Somersalo et al., 2012).

Figure 2*B* shows a representative pH$_S$ recording from an $H_2O$-injected or 'control' oocyte during such a solution change (for solution composition, see Table 1). The vertical grey bands in the figure indicate periods during which we withdraw the tip of the extracellular pH microelectrode from its previous position, in which its tip created a small dimple (~40 μm) in the oocyte surface, to a distance of ~300 μm for recalibration in the bECF, which we assume to have a pH of precisely 7.50. We describe the recalibration in the legend of Fig. 2.

Figure 2*C* shows a representative western blot of membrane preparations (see Methods) of oocytes injected with $H_2O$ or cRNA encoding either hAQP1-WT or hAQP1-C189S. The lower band(s) at ~25 kDa presumably represent unglycosylated or core-glycosylated hAQP1 in the endoplasmic reticulum. The upper bands at ~37 kDa represent mature-glycosylated hAQP1, much of which is presumably at the plasma membrane. Assuming that the AQP1 antibody has similar sensitivities for the WT and C189S proteins, we conclude that *Xenopus laevis* oocytes express similar amounts of hAQP1 for both WT and C189S.

Figure 2*D* and *E* show the structures of the two inhibitors that we use in the present study. Unlike HgCl$_2$, which reacts with a cysteine residue to form the product R-S-Hg-Cl, p-chloromercuribenzene sulfonate (Fig. 2*D*) forms the product R-S-Hg-R′, where R′ is the benzenesulfonate moiety of pCMBS. We use pCMBS rather than HgCl$_2$ because, in our hands, pCMBS is far less toxic to the cells that we study. The inhibitor 4,4′-diisothiocyanatostilbene-2,2′-disulfonate (Fig. 2*E*) has two isothiocyano groups, both of which can react with the deprotonated form of an amino group ($R-NH_3^+ \rightarrow R-NH_2 + H^+$), presumably the $\varepsilon$-amino group of lysine, to form an N,N-disubstituted thiourea $R-NH-CS-NH-R'$, where this R′ is the remainder of the DIDS molecule. Because DIDS is bivalent, it can potentially crosslink two lysine groups. Note that both pCMBS and DIDS are impermeant (Zhao et al., 2025).

Figure 3 shows a series of 12 representative pH$_S$ recordings, the first of which, in Fig. 3*A*, is a replicate of Fig. 2*B*. We can see that the $\Delta pH_S$ induced by the $CO_2$ influx in this $H_2O$-injected control oocyte is much smaller (upward orange arrow in Fig. 3*A*) than in a day-matched oocyte injected with cRNA encoding hAQP1-WT (upward green arrow in Fig. 3*B*). For a larger number of similar experiments, Statistics Table 3 shows that the mean $\Delta pH_S$ induced by the $CO_2$ influx into $H_2O$-injected oocytes (as in Fig. 3*A*) is significantly different from the $CO_2$-induced $\Delta pH_S$ recorded from oocytes expressing hAQP1 (as in Fig. 3*B*).

We define $(\Delta pH_S^*)_{CO2}$ as the difference between the $CO_2$-induced $\Delta pH_S$ value of an oocyte expressing a channel protein (upward black arrow in inset of Fig. 3*B*) and the mean $\Delta pH_S$ for all day-matched control oocytes (e.g., Fig. 3*A*). This difference is a semiquantitative index of the channel-specific $CO_2$ flux (Calvetti et al., 2020; Musa-Aziz, Chen et al., 2009; Somersalo et al., 2012). The leftmost bar in Fig. 4*A* represents the mean $(\Delta pH_S^*)_{CO2}$ value for 34 oocytes expressing hAQP1-WT in the absence of inhibitors.

To assess the importance of the monomeric pore, we examined the effect of pCMBS, which, similar to HgCl$_2$ (Preston et al., 1993), covalently modifies Cys-189 near the extracellular entrance of the monomeric pore and reduces the component of $P_f$ due to AQP1, which we define as $P_f^*$. Preston et al showed that HgCl$_2$ has no effect on $P_f$ if oocytes express the C189S mutant of AQP1 (Preston et al., 1993). In Fig. 3*C*, we show that a representative $\Delta pH_S$ induced by the $CO_2$ influx into an oocyte expressing hAQP1-C189S is substantially larger than that from the $H_2O$ oocyte shown in Fig. 3*A*. For a larger number of similar experiments, Statistics Table 3 shows that the mean $\Delta pH_S$ of oocytes expressing hAQP1-C189S is significantly different from that of $H_2O$-injected oocytes but not significantly different from oocytes expressing hAQP1-WT. On the basis of pH$_i$

**Statistics Table 3. Analysis of the mean pH$_S$ transients triggered by a CO$_2$/HCO$_3^-$ exposure and subsequent CO$_2$ influx from the 'control' H$_2$O, hAQP1 and hAQP1-C189S populations presented in Figure 3*A–C*. The first four columns display from left to right the cRNA injected, Mean $\Delta$pH$_S$ amplitude, standard deviation (S.D.) and number of replicates (*n*) are presented for each type of cRNA injection. The right half of the table presents the one-way ANOVA with Holm-Bonferroni post-hoc means comparisons for each group. FWER $\alpha$ set at 0.05. This half of the table is split into two halves, with the upper-right half showing the adjusted $\alpha$-value for each comparison and the lower-left half the *P*-values. Significant *P*-values are in bold.**

| cRNA | $\Delta$pH$_S$ | S.D. | *n* | *P* \ $\alpha$ | H$_2$O | hAQP1-WT $\alpha$ | hAQP1-C189S $\alpha$ |
|---|---|---|---|---|---|---|---|
| H$_2$O | 0.051 | 0.019 | 23 | | | 0.0167 | 0.0250 |
| hAQP1-WT | 0.116 | 0.034 | 34 | *P* | **1.58$\times$10$^{-12}$** | | 0.0500 |
| hAQP1-C189S | 0.106 | 0.029 | 23 | *P* | **6.96$\times$10$^{-9}$** | 0.194 | |

measurements, Cooper and Boron (1998) showed that pCMBS significantly reduces the CO$_2$ permeability of oocytes expressing hAQP1-WT, but that this effect is absent in oocytes expressing hAQP1-C189S. In the present study, in which we now monitor pH$_S$, experiments on individual oocytes confirm that pretreatment with 1 mM pCMBS (see Methods) reduces the CO$_2$-induced $\Delta$pH$_S$ in

an hAQP1-WT oocyte (Fig. 3*B* vs. *E*) but not in an oocyte expressing hAQP1-C189S (Fig. 3*C* vs. *F*). pCMBS has no effect on an H$_2$O-injected oocyte (Fig. 3*A* vs. *D*), for which $\Delta$pH$_S$ is already small. We conclude that pCMBS reduces but does not eliminate the $\Delta$pH$_S$ due to hAQP1 (Fig. 3D vs. *E*).

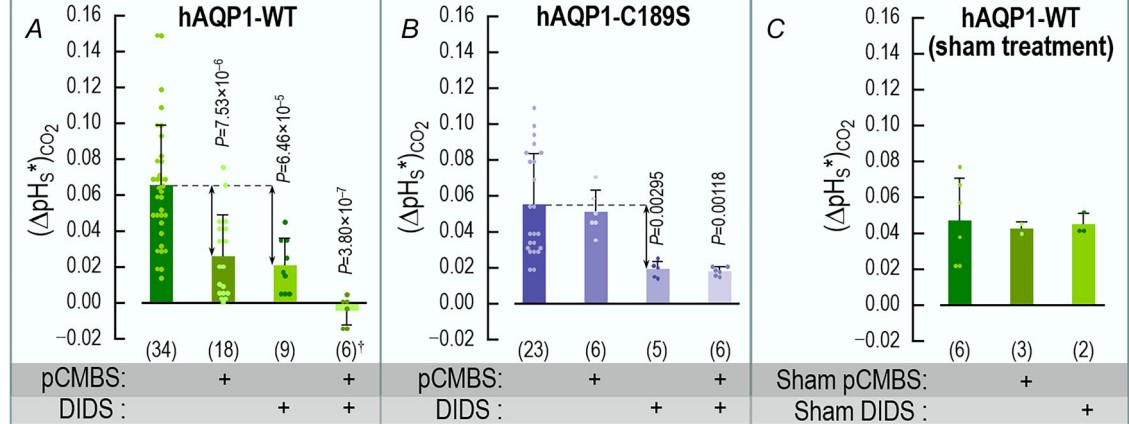

**Figure 4. Summary of channel-specific data in assays for CO$_2$ influx**
A, oocytes expressing hAQP1-WT. pCMBS and DIDS each reduce ($\Delta$pH$_S$*)$_{CO2}$ by somewhat more than half, and the two drugs together reduce it to virtually zero. For the rightmost bar, the dagger symbol (†) adjacent to the replicates (i.e., *n* = 6) indicates that the ($\Delta$pH$_S$*)$_{CO2}$ for the pCMBS+DIDS condition is not significantly different from zero (*P* = 0.254, one-sample *t*-test). Of the 6 pCMBS+DIDS oocytes, 2 were treated first with pCMBS, and then DIDS; the other 4 were in the opposite order. B, oocytes expressing the C189S mutant of hAQP1. DIDS reduces ($\Delta$pH$_S$*)$_{CO2}$ by about half, but pCMBS is without effect, ±DIDS. Of the 6 pCMBS+DIDS oocytes (far-right bar), 4 we treated first with pCMBS, and then DIDS; the other 2 were in the opposite order. C, oocytes expressing hAQP1 but undergoing only sham drug exposures. These sham experiments show that the long protocols did not have a substantial effect on ($\Delta$pH$_S$*)$_{CO2}$. The pCMBS shams (30 min) and DIDS shams (60 min) differed only in the ND96 incubation time in Fig. 1/F$_3$. The data come from experiments like those in Fig. 3, in which we exposed oocytes to 5% CO$_2$/33 mM HCO$_3^-$ (for protocol, see Methods and Fig. 1). From each $\Delta$pH$_S$ of a channel-expressing oocyte (e.g., Fig. 3*B*, *E*, *H*, *K* or *C*, *F*, *I*, *L*), we subtract the mean, day-matched $\Delta$pH$_S$ for the corresponding H$_2$O-injected oocytes (e.g., Fig. 3*A*, *D*, *G* and *J*) to calculate the channel-dependent $\Delta$pH$_S$ for CO$_2$, that is, ($\Delta$pH$_S$*)$_{CO2}$. The ($\Delta$pH$_S$*)$_{CO2}$ values from individual oocytes are plotted as dots over the green-shaded bars in panels A and C, and purple-shaded bars in panel B. At the base of each bar in parentheses is the number of oocytes (*n*), which come from a minimum of 5 batches of oocytes (i.e., different frogs; N $\geq$ 5). Error bars represent S.D. In the horizontal grey bands at the bottom of each panel, '+' indicates a pre-incubation with pCMBS or DIDS, or a sham exposure. *P*-values denote statistically significant differences from the no-drug condition, and are results of one-way ANOVAs amongst all groups, followed by Holm–Bonferroni post hoc means comparisons (see Statistics in Methods). For clarity, we display only *P*-values that indicate statistical significance; we show the *P*-values for all comparisons in Statistics Table 4A, Statistics Table 4B and Statistics Table 4C.

**Statistics Table 4. Tables of *P*-values for one-way ANOVA with Holm-Bonferroni post-hoc means comparison for comparisons of differences of channel-corrected $(\Delta pH_S^*)_{CO2}$ on exposure of the oocytes to 5% $CO_2$/33 mM $HCO_3^-$. Each table is split into two halves, with FWER $\alpha$ set at 0.05, the upper-right half shows the adjusted $\alpha$-value for each comparison and the lower-left half the *P*-value. Significant *P*-values are highlighted in bold. 4A, *Statistics summary of channel-specific data in assays for $CO_2$-influx into hAQP1 expressing oocytes treated with pCMBS, DIDS or pCMBS and DIDS in* Figure 4A. 4B, *Statistics summary of channel-specific data in assays for $CO_2$-influx into hAQP1-C189S expressing oocytes treated with pCMBS, DIDS or pCMBS and DIDS in* Figure 4B. 4C, *Statistics summary of channel-specific data in assays for $CO_2$-influx into hAQP1 expressing oocytes incubated with sham pCMBS or sham DIDS treatments in* Figure 4C**

| hAQP1-WT | | Ctrl | +pCMBS | +DIDS | +pCMBS & +DIDS |
|---|---|---|---|---|---|
| | $P \setminus \alpha$ | | $\alpha$ | $\alpha$ | $\alpha$ |
| Ctrl | | | 0.0100 | 0.0125 | 0.00833 |
| +pCMBS | *P* | **7.53×10⁻⁶** | | 0.0500 | 0.0167 |
| +DIDS | *P* | **6.46×10⁻⁵** | 0.657 | | 0.0250 |
| +pCMBS & +DIDS | *P* | **3.80×10⁻⁷** | 0.0242 | 0.0902 | |
| hAQP1-C189S | | Ctrl | +pCMBS | +DIDS | +pCMBS & +DIDS |
| | $P \setminus \alpha$ | | $\alpha$ | $\alpha$ | $\alpha$ |
| Ctrl | | | 0.0250 | 0.0100 | 0.00833 |
| +pCMBS | *P* | 0.696 | | 0.0167 | 0.0125 |
| +DIDS | *P* | **0.00295** | 0.0273 | | 0.0500 |
| +pCMBS & +DIDS | *P* | **0.00118** | 0.0177 | 0.946 | |
| hAQP1-WT | | Ctrl | Sham +pCMBS | | Sham +DIDS |
| | $P \setminus \alpha$ | | $\alpha$ | | $\alpha$ |
| Ctrl | | | 0.0167 | | 0.0250 |
| Sham +pCMBS | *P* | 0.768 | | | 0.0500 |
| Sham +DIDS | *P* | 0.874 | 0.887 | | |

The leftmost bar in Fig. 4*B* represents the mean $(\Delta pH_S^*)_{CO2}$ from 22 oocytes expressing hAQP1-C189S in the absence of inhibitors. This $(\Delta pH_S^*)_{CO2}$ value is very similar to that for oocytes expressing hAQP1-WT (leftmost bar in Fig. 4*A*). A comparison of the first and second bars in Fig. 4*A* and *B* shows that pCMBS reduces $(\Delta pH_S^*)_{CO2}$ by somewhat more than half in oocytes expressing hAQP1-WT, but has no significant effect in hAQP1-C189S oocytes. These data support the hypothesis that, of the $CO_2$ that transits hAQP1, a major component moves through the same monomeric pores as $H_2O$.

Although studying $HCO_3^-$ transport, Forster et al unexpectedly found that DIDS blocks a large fraction of the $CO_2$ permeability of human RBCs (Forster et al., 1998). Endeward et al found that, in RBCs genetically deficient in hAQP1, DIDS has a reduced effect on $CO_2$ permeability (Endeward et al., 2006). Here, in the present experiments on single oocytes, we find that pretreatment with 100 µM DIDS reduces the $CO_2$-induced $\Delta pH_S$ in both an hAQP1-WT oocyte (Fig. 3*H* vs. *B*) and an hAQP1-C189S oocyte (Fig. 3*I* vs. *C*), but not in a $H_2O$-injected oocyte (Fig. 3*G* vs. *A*), for which $\Delta pH_S$ is already small.

The third bars in Fig. 4*A* and *B* summarise mean $(\Delta pH_S^*)_{CO2}$ values from a larger group of oocytes pretreated with DIDS. A comparison of the first and third bars shows that DIDS reduces $(\Delta pH_S^*)_{CO2}$ by more than half in oocytes expressing either hAQP1-WT (Fig. 4*A*) or hAQP1-C189S (Fig. 4*B*). These data are consistent with the hypothesis that a significant fraction of $CO_2$ moves through hAQP1 by a DIDS-sensitive pathway that is unaffected by the C189S mutation in the extracellular mouth of the monomeric pore.

Returning to experiments on individual oocytes, we see that the sequential exposure of an hAQP1-WT oocyte to pCMBS and then DIDS substantially reduces the $CO_2$-induced $\Delta pH_S$ (Fig. 3*K* vs. *B*). Moreover, the size of the $\Delta pH_S$ for the pCMBS/DIDS-treated hAQP1-WT oocyte (Fig. 3*K*) is about the same as for $H_2O$-injected oocytes ± inhibitors (Fig. 3*A*,*D*,*G* and *J*). In an hAQP1-C189S oocyte, the combination of pretreating with pCMBS then DIDS reduces the $CO_2$-induced $\Delta pH_S$ (Fig. 3*L* vs. *C*), but by no more than DIDS alone (Fig. 3*I* vs. *C*).

The fourth bars in Fig. 4*A* and *B* summarise mean $(\Delta pH_S^*)_{CO2}$ values from a larger group of oocytes, in experiments in which we treated with pCMBS and DIDS in either order. A comparison of the first and fourth bars shows that pCMBS+DIDS reduces $(\Delta pH_S^*)_{CO2}$ by slightly more than 100% in hAQP1-WT oocytes (Fig. 4*A*). In the hAQP1-C189S oocytes, pCMBS+DIDS reduces $(\Delta pH_S^*)_{CO2}$ to about the same extent as DIDS alone (Fig. 4*B*), about 40%. Taken together, the data in Figs 3 and 4*A*, *B* are consistent with the hypothesis that, in addition to the component of $CO_2$ that moves through the four monomeric pores, another component—at

least as large—moves through an entirely separate, DIDS-sensitive pathway.

Figure 4*C* summarises the results of ND96 sham experiments in which—in step F$_3$ of Fig. 1—we simulated either a 30-min pCMBS exposure ($n = 3$) or a 60-min DIDS exposure ($n = 2$). The averages of the two sham groups were nearly identical to those of the control group, showing that the oocytes can tolerate these long protocols.

**pH$_S$ measurements for NH$_3$ transport.** Figure 5*A* is a schematic representation of the reaction and diffusion events that take place as we expose an oocyte to a solution containing 0.5 mM NH$_3$/NH$_4$$^+$. Here, the changes in pH$_S$ are opposite in direction to those caused by CO$_2$/HCO$_3$$^-$ exposure. As the weak base NH$_3$ enters the cell, it causes a decrease in [NH$_3$]$_S$, which has two effects. First, it provides a gradient for NH$_3$ diffusion from the bECF to the cell surface. By itself, this diffusion, which partially replenishes NH$_3$ at the cell surface, has no effect on pH. Second, the decrease in [NH$_3$]$_S$ also drives the reaction NH$_4$$^+$→NH$_3$+H$^+$ at the cell surface, which also partially replenishes NH$_3$. This reaction produces an acidic pH$_S$ transient (Musa-Aziz et al., 2009; Musa-Aziz, Chen et al., 2009), so that $\Delta$pH$_S$<0. Figure 5*B* shows a representative pH$_S$ recording as we introduce NH$_3$/NH$_4$$^+$.

Figure 6 shows a series of 9 representative pH$_S$ recordings on individual oocytes, the first of which, in Fig. 6*A*, is a replicate of Fig. 5*B*. We find that

the NH$_3$-induced '−$\Delta$pH$_S$' is much smaller in the H$_2$O-injected oocyte (Fig. 6*A*) than in a day-matched oocyte expressing either hAQP1-WT (Fig. 6*B*) or hAQP1-C189S (Fig. 6*C*). For a larger number of similar experiments, Statistics Table 6 shows that the mean $\Delta$pH$_S$ induced by NH$_3$ influx into H$_2$O-injected oocytes (as in Fig. 6*A*) is significantly different from the NH$_3$-induced $\Delta$pH$_S$ recorded from oocytes expressing either hAQP1-WT (as in Fig. 6*B*) or hAQP1-C189S (as in Fig. 6*C*); however, the $\Delta$pH$_S$ values for oocytes expressing the two hAQP1 constructs were not significantly different. Subtracting the $\Delta$pH$_S$ from the H$_2$O-injected oocyte from the $\Delta$pH$_S$ from the hAQP1-WT oocyte yields the channel-dependent, NH$_3$-induced $\Delta$pH$_S$—($\Delta$pH$_S$*)$_{NH3}$ (Musa-Aziz, Chen et al., 2009). The leftmost bar in Fig. 7*A* represents the mean ($\Delta$pH$_S$*)$_{NH3}$ for 18 oocytes expressing hAQP1-WT in the absence of inhibitors. In experiments on representative oocytes, we find that pretreating an hAQP1-WT oocyte with 1 mM pCMBS markedly reduces '−$\Delta$pH$_S$' (Fig. 6*E* vs. *B*), but has no effect in an hAQP1-C189S oocyte, where '−$\Delta$pH$_S$' remains high (Fig. 6*F* vs. *C*); or in an H$_2$O oocyte, where '−$\Delta$pH$_S$' remains low (Fig. 6*D* vs. *A*). The leftmost bar in Fig. 7*B* represents the mean ($\Delta$pH$_S$*)$_{NH3}$ value from 12 oocytes expressing hAQP1-C189S in the absence of inhibitors. This ($\Delta$pH$_S$*)$_{NH3}$ value is virtually identical to that for oocytes expressing hAQP1-WT (leftmost bar in Fig. 7*A*). A comparison of the first and second bars show

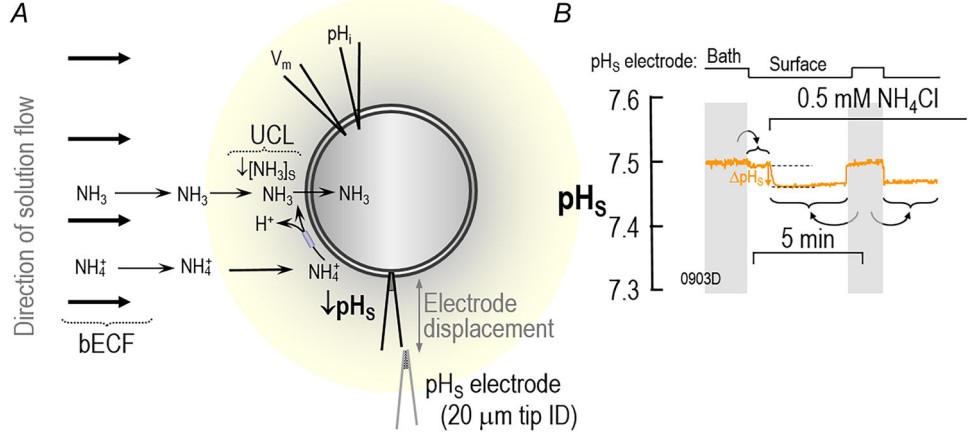

**Figure 5. NH$_3$ influx into an oocyte.**
A, schematic illustration of a NH$_3$-influx experiment. Analogous to Fig. 2, the thick black arrows indicate the direction of convective flow within the bulk extracellular fluid (bECF). Thinner arrows indicate solute diffusion or reactions. The maize-coloured halo indicates the layer of extracellular unconvected fluid (EUF). At the outer surface of the membrane, NH$_3$ influx creates an NH$_3$ deficit, in part replenished by the reaction NH$_4$$^+$ → NH$_3$ + H$^+$ (which decreases pH$_S$) and in part replenished by diffusion from the bECF (an isohydric process). The double-headed arrow indicates movement of the pH$_S$ electrode from the bECF to the cell surface for recalibration. B, Example of a surface-pH (pH$_S$) record. The downward arrow indicates the maximal change in pH$_S$ ($\Delta$pH$_S$) during the application of NH$_3$/NH$_4$$^+$. During the two periods indicated by the vertical grey bands, the tip of the pH$_S$ electrode was withdrawn ~300 μm away from the cell surface for calibration in the bulk extracellular fluid (i.e., pH = 7.50). The curved grey arrows point to horizontal braces that indicate the portions of the experiment to which each calibration pertains. ID, inner diameter. The filename for this representative trace, '0903D'—that we reuse in Fig. 6*A*—is annotated at the bottom left corner of the panel.

**Statistics Table 6. Analysis of the mean $pH_S$ transients triggered by a 0.5 mM $NH_4Cl$ exposure from the 'control' $H_2O$, hAQP1-WT and hAQP1-C189S populations presented in Figure 6*A*, *B* and *C*. The first four columns display, from left to right, the cRNA injected, the mean $\Delta pH_S$ amplitude, standard deviation (S.D.), and number of replicates (n) presented for each type of cRNA injection. The right-half of the table presents the one-way ANOVA with Holm-Bonferroni post-hoc means comparisons for each group. FWER $\alpha$ set at 0.05. This half of the table is split into two halves, with the upper-right half showing the adjusted $\alpha$-value for each comparison and the lower-left half the *P*-value. Significant *P*-values are highlighted bold.**

| cRNA | $\Delta pH_S$ | S.D. | n | $P \backslash \alpha$ | $H_2O$ $\alpha$ | hAQP1-WT $\alpha$ | hAQP1-C189S $\alpha$ |
|---|---|---|---|---|---|---|---|
| $H_2O$ | 0.0387 | 0.013 | 25 | | | 0.0167 | 0.0250 |
| hAQP1-WT | 0.0618 | 0.011 | 18 | *P* | **$6.77 \times 10^{-8}$** | | 0.0500 |
| hAQP1-C189S | 0.0600 | 0.009 | 12 | *P* | **$4.74 \times 10^{-6}$** | 0.689 | |

**Statistics Table 7. Tables of *P*-values for one-way ANOVA with Holm-Bonferroni post-hoc means comparison for channel corrected $(\Delta pH_S*)_{NH3}$ during exposure of the oocytes to 0.5 mM $NH_4Cl$. Each table is split into two halves by the black-shaded cells, with FWER $\alpha$ set at 0.05, the upper-right half shows the adjusted $\alpha$-value for each comparison and the lower-left half the *P*-value. Significant *P*-values are highlighted bold. 7A, Statistics summary of channel-specific data in assays for $NH_3$-influx into hAQP1 expressing oocytes treated with pCMBS, or DIDS in Figure 7A. 7B, Statistics summary of channel-specific data in assays for $NH_3$-influx into hAQP1-C189S expressing oocytes treated with pCMBS, or DIDS in Figure 7B.**

| hAQP1-WT | $P \backslash \alpha$ | Ctrl | +pCMBS $\alpha$ | +DIDS $\alpha$ |
|---|---|---|---|---|
| Ctrl | | | 0.0250 | 0.0500 |
| +pCMBS | *P* | $6.34 \times 10{-5}$ | | 0.0167 |
| +DIDS | *P* | 0.551 | $4.38 \times 10{-5}$ | |
| hAQP1-C189S | $P \backslash \alpha$ | Ctrl | +pCMBS $\alpha$ | +DIDS $\alpha$ |
| Ctrl | | | 0.0167 | 0.0500 |
| +pCMBS | *P* | 0.636 | | 0.0250 |
| +DIDS | *P* | 0.984 | 0.694 | |

**Statistics Table 8. for Figure 8 Tables of *P*-values for one-way ANOVA with Holm-Bonferroni post-hoc means comparison for channel corrected $P_f*$ after incubation in no drug control (Ctrl), +pCMBS or +DIDS solutions. Ctrl oocytes are separated into two groups. Those from the same oocyte preparations (frogs) pre-incubated with pCMBS and those from the same oocyte preparations pre-incubated with DIDS. Each table is split into two halves by the black-shaded cells, with FWER $\alpha$ set at 0.05, the upper-right half shows the adjusted $\alpha$-value for each comparison, and the lower-left half the *P*-value. Significant *P*-values are highlighted in bold. Cells shaded grey are comparisons between conditions performed on oocytes from different frogs; therefore, the comparisons are not pertinent even if the *P*-value is statistically significant. 8A, Statistics summary channel-specific $P_f$ data in assays for osmotic water permeability for hAQP1 oocytes in Figure 8A. 8B, summary channel-specific $P_f$ data in assays for osmotic water permeability for hAQP1-C189S oocytes in Figure 8B**

| hAQP1-WT | $P \backslash \alpha$ | Ctrl for +pCMBS | +pCMBS $\alpha$ | Ctrl for +DIDS $\alpha$ | +DIDS $\alpha$ |
|---|---|---|---|---|---|
| Ctrl for +pCMBS | | | 0.00833 | 0.0250 | 0.0167 |
| +pCMBS | *P* | **$1.70 \times 10^{-12}$** | | 0.0100 | 0.0125 |
| Ctrl for +DIDS | *P* | 0.0632 | **$9.02 \times 10^{-7}$** | | 0.0500 |
| +DIDS | *P* | **0.00761** | **$8.72 \times 10^{-5}$** | 0.395 | |
| hAQP1-C189S | $P \backslash \alpha$ | Ctrl for +pCMBS | +pCMBS $\alpha$ | Ctrl for +DIDS $\alpha$ | +DIDS $\alpha$ |
| Ctrl for +pCMBS | | | 0.0167 | 0.0125 | 0.0500 |
| +pCMBS | *P* | 0.514 | | 0.0100 | 0.0250 |
| Ctrl for +DIDS | *P* | 0.171 | 0.0644 | | 0.00833 |
| +DIDS | *P* | 0.722 | 0.702 | 0.0619 | |

that pCMBS reduces '$-(\Delta pH_S^*)_{NH3}$' by virtually 100% in oocytes expressing hAQP1-WT (Fig. 7*A*), but has no effect in hAQP1-C189S oocytes (Fig. 7*B*).

Returning to experiments on individual oocytes, we see that DIDS has no effect on '$-\Delta pH_S$' in an H$_2$O-injected control oocyte (Fig. 6*G* vs. *A*), where '$-\Delta pH_S$' remains low; or in an hAQP1-WT oocyte (Fig. 6*H* vs. *B*) or an hAQP1-C189S oocyte (Fig. 6*I* vs. *C*), where '$-\Delta pH_S$' remains high. The third bars in Fig. 7*A* and *B* summarise mean $(\Delta pH_S^*)_{NH3}$ values from larger groups of DIDS-pretreated oocytes. A comparison of the first and third bars in each panel shows that DIDS is without effect on $(\Delta pH_S^*)_{NH3}$ in oocytes expressing hAQP1-WT (Fig. 7*A*) or hAQP1-C189S (Fig. 7*B*). Taken together, the data in Figs 6 and 7 indicate that all of the NH$_3$ transiting hAQP1 moves via the same pathway as H$_2$O and one of the two major components of CO$_2$—namely the four pCMBS-sensitive monomeric pores. None of the NH$_3$ moves via the alternate, DIDS-sensitive pathway taken by the other major component of CO$_2$.

**Cell-swelling experiments.** In parallel with the CO$_2$ and NH$_3$ assays (Figs 4 and 7), we determined $P_f^*$. In Fig. 8*A*, we summarise our hAQP1-WT data, which confirm earlier work on RBCs and oocytes heterologously expressing AQP1, mercurials reduce $P_f^*$ by about half (Kabutomori et al., 2018; Macey, 1984; Musa-Aziz, Chen et al., 2009; Preston et al., 1993; Zeidel et al., 1994) but that DIDS is without effect (Endeward et al., 2006; Macey, 1984). Figure 8*B* confirms that the hAQP1-C189S mutation renders $P_f^*$ insensitive to pCMBS (Cooper & Boron, 1998; Kabutomori et al., 2018), and also shows that the mutation does not affect the lack of DIDS sensitivity.

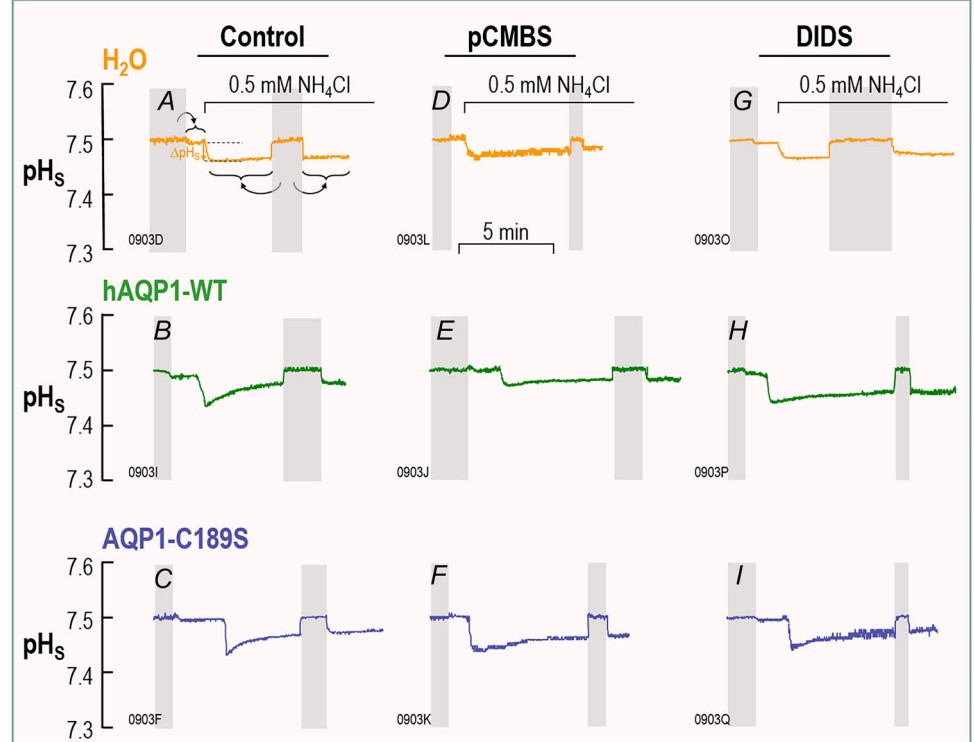

**Figure 6. Surface-pH transients triggered by NH$_3$/NH$_4^+$ exposure and the subsequent NH$_3$ influx**
The figure shows representative pH$_S$ records from oocytes injected with H$_2$O (top row) or cRNA encoding hAQP1-WT (middle row) or its C189S mutant (bottom row), and later exposed first to the ND96 solution and then, at the indicated times, to a solution containing NH$_3$/NH$_4^+$. All oocytes underwent the 'Control' protocol, followed by the F$_1$ or F$_2$ protocol in Fig. 1, although some oocytes did not survive beyond the control protocol (see Methods). Statistics Table 6 presents the analysis of the mean $\Delta pH_S$ magnitude recorded from all H$_2$O, hAQP1 or hAQP1-C189 'control' oocytes for which we show representative traces in panels *A*, *B* and *C*. As necessary, we pre-incubated oocytes in 1 mM pCMBS for 30 min, or in 100 μM DIDS for 1b h. Neither drug was present in the bulk solution at the time of the assays. The grey bars indicate when we withdrew the pH-electrode tip from the oocyte surface to the bulk extracellular solution (pH 7.50) for recalibration. In each panel, the filename of the representative trace is annotated at the bottom left corner of the panel. '0903D' in panel A is a reproduction of the recording in Fig. 5*B*.

**Statistics Table 9.** for Figure 9 Tables of *P*-values for one-way ANOVA with Holm-Bonferroni post-hoc means comparison for channel corrected A, $(\Delta pH_S^*)_{CO2}$ in hAQP1-WT oocytes, B, $(\Delta pH_S^*)_{CO2}$ in hAQP1-FLAG tagged oocytes, C, $P_f^*$ in hAQP1-WT oocytes and D, $P_f^*$ in hAQP1-FLAG tagged oocytes. Each table is split in two halves by the black-shaded cells, with FWER $\alpha$ set at 0.05, the upper-right half shows the adjusted $\alpha$-value for each comparison and the lower-left half the *P*-value. Significant *P*-values are highlighted bold. 9A, Statistics summary of channel-specific $CO_2$ influx data for into hAQP1-WT oocytes. 9B, Statistics summary of channel-specific $CO_2$ influx data for into hAQP1-FLAG oocytes. 9C, Statistics summary of channel-specific $P_f$ data for into oocytes expressing hAQP1-WT. 9D, Statistics summary of channel-specific $P_f$ data for into hAQP1-FLAG oocytes.

| hAQP1-WT | | Ctrl | +DIDS | +DIDS +Alb 0.2% |
|---|---|---|---|---|
| | $P \setminus \alpha$ | | $\alpha$ | $\alpha$ |
| Ctrl | | | 0.0250 | 0.0167 |
| +DIDS | $P$ | **0.00976** | | 0.0500 |
| +DIDS +Alb 0.2% | $P$ | **0.00682** | 0.955 | |
| hAQP1-FLAG | | Ctrl | +DIDS | +DIDS +Alb 0.2% |
| | $P \setminus \alpha$ | | $\alpha$ | $\alpha$ |
| Ctrl | | | 0.0250 | 0.0167 |
| +DIDS | $P$ | **$3.01 \times 10^{-5}$** | | 0.0500 |
| +DIDS +Alb 0.2% | $P$ | **$4.05 \times 10^{-6}$** | 0.740 | |
| AQP1 | | Ctrl | +pCMBS | +DIDS |
| | $P \setminus \alpha$ | | $\alpha$ | $\alpha$ |
| Ctrl | | | 0.0167 | 0.0500 |
| +pCMBS | $P$ | **$4.73 \times 10^{-5}$** | | 0.0250 |
| +DIDS | $P$ | 0.361 | **0.00483** | |
| hAQP1-FLAG | | Ctrl | +pCMBS | +DIDS |
| | $P \setminus \alpha$ | | $\alpha$ | $\alpha$ |
| Ctrl | | | 0.0167 | 0.0500 |
| +pCMBS | $P$ | **$7.49 \times 10^{-8}$** | | 0.0250 |
| +DIDS | $P$ | 0.322 | **$1.25 \times 10^{-4}$** | |

## Modification of hAQP1 by DIDS

Via rapid and reversible electrostatic interactions, the two sulfonate groups of DIDS can reversibly interact with cationic sites on proteins. Via slower covalent reactions with the $-NH_2$ group of lysine or $-OH/-SH$ groups of other amino acids, the two isothiocyanate groups of DIDS can act as a homobifunctional crosslinking reagent (Cabantchik & Rothstein, 1972; Lepke et al., 1976). The pair of sulfonate groups on DIDS endows this divalent anion with low membrane permeability.

**$pH_S$ experiments on oocytes.** To assess whether the DIDS inhibition occurs via an electrostatic or covalent interaction, we followed a DIDS exposure with an albumin wash (see Methods) to scavenge non-covalently bound DIDS. Figure 9A shows that the albumin wash does not significantly reduce the degree to which the DIDS pretreatment decreases $(\Delta pH_S^*)_{CO2}$. Because the inhibition is irreversible, the interaction of the DIDS with hAQP1 is most likely covalent. To investigate this covalent modification, we performed the biochemistry experiments (presented in the next section) in which we FLAG-tagged hAQP1 at its N-terminus, overexpressed the construct in *Pichia pastoris*, and prepared spheroplasts. Here, in control oocyte experiments, we observe that the FLAG tag has no effect on the $(\Delta pH_S^*)_{CO2}$ $\pm$DIDS or $\pm$albumin (Fig. 9A vs. B). We also find that the FLAG tag is without effect in $P_f$ assays, $\pm$pCMBS or $\pm$DIDS (Fig. 9C vs. D).

**Biochemistry experiments.** In separate studies, as mentioned in the previous paragraph, we overexpressed FLAG-tagged hAQP1 in *Pichia pastoris* and prepared spheroplasts. We treated intact spheroplasts with the membrane-impermeant DIDS (or, as shams, without DIDS), solubilised the membranes in detergent, and purified hAQP1 using an anti-FLAG resin. Further purification by size-exclusion chromatography (Fig. 10A) shows that DIDS treatment promotes the formation of higher–molecular-weight hAQP1 species, as indicated by the increased peak height of the void volume ($V_0$). We then pooled and concentrated fractions from each peak. Optical spectroscopy of this material (Fig. 10B) reveals the expected increase in absorbance centered at ~340 nm (brown arrow; see Kodippili et al., 2009), due to DIDS in the DIDS-treated vs. sham samples. We also separated the pooled/concentrated material by SDS-PAGE and transferred it to membranes for western blotting.

In the anti-FLAG blot (Fig. 10C), the dominant species are hAQP1 monomers ($\leq$50 kDa, arrows a–c). Band a likely represents a C-terminal cleavage product, and band b probably reflects unglycosylated or core-glycosylated

protein. Thus, both presumably represent mainly intracellular proteins (i.e., not on the plasma membrane). Band c likely represents mature-glycosylated monomers, some of which may be tetramers that had trafficked to the plasma membrane but dissociated during the preparative procedure. It is perhaps noteworthy that band c is more intense in cells treated with DIDS. Presumed dimers (arrow d) and tetramers (arrow e) are visible only in DIDS-treated samples (lanes 2, 4, 6), and may represent proteins accessible to DIDS at the outer cell surface. In preliminary work on hAQP1 expressed in oocytes, we similarly found that DIDS markedly increases the appearance of dimers (Geyer et al., 2011).

In the anti-DIDS blot (Fig. 10*D*), the monomeric species expected to traffic poorly to the plasma membrane (arrows a–b, but dominant in the anti-FLAG blot) are poorly represented. Indeed, we expect cleaved hAQP1 (band a) and hAQP1 less than mature-glycosylated (band b) to be largely inaccessible to the impermeant DIDS.

Band c, the presumed mature-glycosylated monomers, is not apparent in the anti-DIDS blot. We suggest that hAQP1 tetramers at the *Pichia* cell-surface that did not react with DIDS subsequently dissociated during the preparative process—a possible explanation for the greater intensity of band c in the DIDS-treated samples in Fig. 10*C* but their virtual absence in Fig. 10*D*. Finally, the presumed hAQP1 dimers and tetramers are clearly visible, though only in DIDS-treated samples (lanes 2, 4, 6).

These data in Fig. 10 are consistent with the hypothesis that, once the divalent DIDS, with isothiocyano groups at opposite ends of the molecule, reacts to one hAQP1 monomer on the surface of *Pichia* cells, the odds are high that the DIDS crosslinks to one more monomer to yield dimers and tetramers that survive the solubilisation and isolation procedure. Although we attempted to use mass spectrometry to identify hAQP1 residue(s) derivatised by DIDS, we were unable to achieve coverage of protein fragments containing predicted DIDS-reactive sites near hydrophobic transmembrane segments.

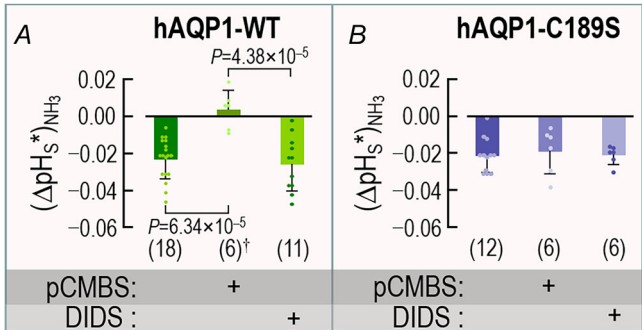

**Figure 7. Summary of channel-specific data in assays for NH₃ influx**

A, oocytes expressing hAQP1-WT. pCMBS reduces $(\Delta pH_S{}^*)_{NH3}$ to virtually zero, whereas DIDS has no effect. For the middle bar, the dagger symbol (†) adjacent to the replicates (i.e., $n = 6$) indicates that the $(\Delta pH_S{}^*)_{NH3}$ for the pCMBS condition is not significantly different from zero ($P = 0.450$, one-sample $t$-test). B, oocytes expressing the C189S mutant of hAQP1. Neither drug affects $(\Delta pH_S{}^*)_{NH3}$. The data come from experiments like those in Fig. 6, in which we exposed oocytes to 0.5 mM NH₄Cl (for protocol, see Methods and Fig. 1). From each $\Delta pH_S$ of a channel-expressing oocyte (Fig. 6*B, E, H* or *C, F, I*), we subtracted the corresponding mean, day-matched $\Delta pH_S$ for H₂O-injected oocytes (Fig. 6*A, D, G*) to calculate the channel-dependent $\Delta pH_S$ for NH₃, that is, $(\Delta pH_S{}^*)_{NH3}$. The $(\Delta pH_S{}^*)_{NH3}$ values from individual oocytes are plotted over the green-shaded bars in panel A, and purple-shaded bars in panel B. At the base of each bar in parentheses is the number of oocytes (*n*), which come from a minimum of 5 batches of oocytes (i.e., different frogs; N). Error bars represent S.D. In the horizontal grey bands at the bottom of each panel, '+mean ' indicates a pre-incubation with pCMBS or DIDS. *P*-values denote statistically significant differences from the no-drug condition, and are the results of one-way ANOVAs amongst all groups, followed by Holm–Bonferroni post-hoc means comparisons (see Statistics in Methods). For clarity, we display only *P*-values that indicate statistical significance; we show *P*-values for all comparisons in Statistics Table 7*A* and Statistics Table 7B.

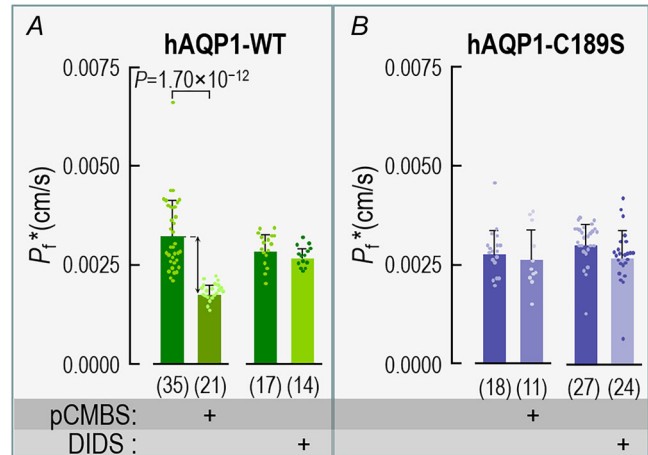

**Figure 8. Summary of channel-specific data in assays for osmotic water permeability**

A, oocytes expressing hAQP1-WT. pCMBS but not DIDS reduces $P_f$. B) oocytes expressing the C189S mutant of hAQP1. Neither drug reduces $P_f$. From each $P_f$ from a channel-expressing oocyte, we subtracted the corresponding mean, day-matched $P_f$ for H₂O-injected oocytes to calculate the channel-dependent $P_f$, that is, $P_f{}^*$. The $P_f{}^*$ values from individual oocytes are plotted over the green-shaded bars in panel A, and purple-shaded bars in panel B. At the base of each bar in parentheses is the number of oocytes (*n*), which come from a minimum of 5 batches of oocytes (i.e., different frogs; N). Error bars represent S.D. In the horizontal grey bands at the bottom of each panel, '+' indicates a pre-incubation with pCMBS or DIDS. *P*-values denote statistically significant differences from the no-drug condition, and are the results of a one-way ANOVA amongst all groups, followed by Holm–Bonferroni post-hoc means comparisons (see Statistics in Methods). For clarity, we display only *P*-values that indicate statistical significance; we show *P*-values for all comparisons in Statistics Table 8.

## Discussion

### Pathways for $NH_3$ versus $CO_2$

**$NH_3$ permeation through monomeric pores.** Although previous work had shown that hAQP1 can serve as a conduit for $NH_3$ (Nakhoul et al., 2001), the present work is the first direct demonstration that the four mono-

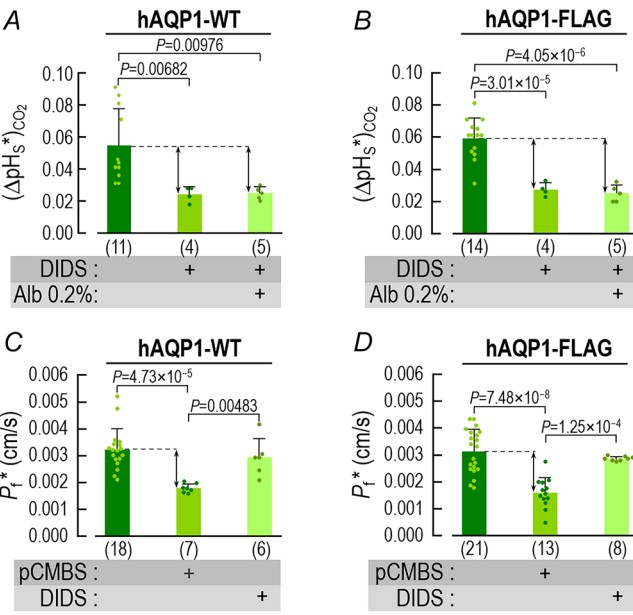

**Figure 9. Summary of channel-specific data for $CO_2$ and $H_2O$ influx for WT versus FLAG-tagged hAQP1**

A, Oocytes expressing hAQP1-WT: Effect of albumin washes on the DIDS sensitivity of $(\Delta pH_S^*)_{CO2}$. DIDS reduces $(\Delta pH_S^*)_{CO2}$ by about half, regardless of the subsequent albumin wash. B, Oocytes expressing N-terminally FLAG-tagged hAQP1: Effect of albumin washes on the DIDS sensitivity of $(\Delta pH_S^*)_{CO2}$. Also, in the FLAG-tagged construct, DIDS reduces $(\Delta pH_S^*)_{CO2}$ by about half, regardless of the albumin wash. C, Oocytes expressing hAQP1-WT: The DIDS sensitivity of $P_f^*$. These experiments are a matched control for the study in panel D. As in the comparable experiments summarised in Figure 8A, pCMBS but not DIDS reduces $P_f$. D, Oocytes expressing FLAG-tagged hAQP1: The DIDS sensitivity of $P_f^*$. As in panel C (no FLAG tag), pCMBS but not DIDS reduces $P_f$ in oocytes expressing FLAG-tagged hAQP1. For panels A and B, the control $pH_S$ data (leftmost dark-green bars) come from experiments like those in Figure 3, whereas for panels C and D, the control $P_f$ data come from experiments like those in Figure 8. The $(\Delta pH_S^*)_{CO2}$ or $P_f^*$ values from individual oocytes are plotted as dots over the green-shaded bars in panels A through D. At the base of each bar in parentheses is the number of oocytes ($n$), which come from a minimum of 5 batches of oocytes (i.e., different frogs; N ≥ 5). Error bars represent S.D. In the horizontal grey bands at the bottom of each panel, '+' indicates a pre-incubation with pCMBS or DIDS, or the presence of 0.2% bovine serum albumin (Alb 0.2%) P-values denote statistically significant differences from the no-drug condition, and are results of one-way ANOVAs amongst all groups, followed by Holm–Bonferroni which means comparisons (see Statistics in Methods). For clarity, we display only the P-values that indicate statistical significance; we show P-values for all comparisons in Statistics Table 9.

meric pores of hAQP1—which are responsible for all $H_2O$ conductance (Preston et al., 1993)—are, in fact, also responsible for all $NH_3$ conductance. This result is not unexpected, given the hydrophilicity of the monomeric pore (Sui et al., 2001; Tajkhorshid et al., 2002; Walz et al., 1997), and the similarities of the electronic structures of the $NH_3$ molecule (i.e., 1 lone pair of electrons+3 N–H bonds amongst 4 $sp^3$ hybrid orbitals) and the $H_2O$ molecule (i.e., 2 lone pairs+2 O–H bonds amongst 4 $sp^3$ hybrid orbitals). Three major lines of evidence support our conclusion that the predominant $NH_3$ pathway through hAQP1 is the monomeric pore. First, $NH_3$ and $H_2O$ have similar chemistries, as just noted. Second, pCMBS reduces $(\Delta pH_S^*)_{NH3}$ to zero (Fig. 7A, centre vs. left bars), an effect abrogated by the C189S mutation (Fig. 7B, analogous bars). Third, DIDS has no effect on either $(\Delta pH_S^*)_{NH3}$ (Fig. 7A and B, right vs. left bars) or $P_f^*$ (Fig. 8A), which leads to a parallel conclusion that virtually none of the $NH_3$ or $H_2O$ moves via the alternate hAQP1 pathway (perhaps the central pore).

Although the above results and interpretations for $NH_3$ permeability parallel those for $H_2O$ permeability, the parallelism is not perfect. As noted in Results[6], pCMBS reduces $(\Delta pH_S^*)_{NH3}$ to zero, but only reduces $P_f^*$ by half (Fig. 8). Thus, although the chemistries governing the movements of $NH_3$ and $H_2O$ through the monomeric pore—including the impact of derivatisation of C189 by Hg-containing compounds—are similar, they are clearly not identical. Finally, as long appreciated from the work of Preston et al. (1993), and confirmed in the present study, derivatisation of C189 does not block all traffic through the monomeric pore of hAQP1.

**$CO_2$ permeation through monomeric pores.** Cooper and Boron (1998) had previously examined the effect of pCMBS on the maximal rate of $pH_i$ decrease—$(dpH_i/dt)_{Max}$—elicited by the introduction of 1.5% $CO_2$/10 mM $HCO_3^-$. They found that, in oocytes expressing hAQP1-WT (but not those expressing hAQP1-C189S or injected with $H_2O$), pCMBS causes a change (a decrease) in $(dpH_i/dt)_{Max}$ that was statistically significant (their Fig. 3D). Their statistical analysis did not address the question of whether pCMBS reduced the hAQP1-dependent component of $(dpH_i/dt)_{Max}$ to zero[7]. Nevertheless, in hAQP1-expressing oocytes, they found that pCMBS lowered $(dpH_i/dt)_{Max}$ nearly to the value observed for $H_2O$-injected control oocytes exposed to pCMBS (their Fig. 3C).

---

[6]See Results > Surface-pH Measurements … > Cell-swelling experiments

[7]It was later that our laboratory introduced the practice of subtracting the background signal from day-matched controls to produce the channel-dependent signal, indicated by the *

The present work shows that, with the same $CO_2/HCO_3^-$ exposure as Cooper, pCMBS causes a change (a decrease) in $(\Delta pH_S^*)_{CO_2}$ that is statistically significant; pCMBS causes $(\Delta pH_S^*)_{CO_2}$ to fall by somewhat more than half. Examination of the Statistics Table 4A shows that, in Fig. 4*A*, the difference between the pCMBS bar and pCMBS+DIDS bar (which we could regard as 100% blockade) falls short of being statistically significant ($\alpha = 0.0100$, $P = 0.0242$). Thus, from a statistical point of view, the present $(\Delta pH_S^*)_{CO_2}$ data

confirm the earlier $(dpH_i/dt)_{Max}$ observations of Cooper & Boron (1998). That pCMBS reduces $(\Delta pH_S^*)_{CO_2}$ by somewhat more than half in the present study but reduced $(dpH_i/dt)_{Max}$ to nearly background in the Cooper study may reflect differences in the precision of the two methods.

We conclude that at least some $CO_2$ moves through hAQP1 via the monomeric pores. Furthermore, on the basis of the present $(\Delta pH_S^*)_{CO_2}$ data that we regard as being more reliable than the earlier $(dpH_i/dt)_{Max}$ data, we

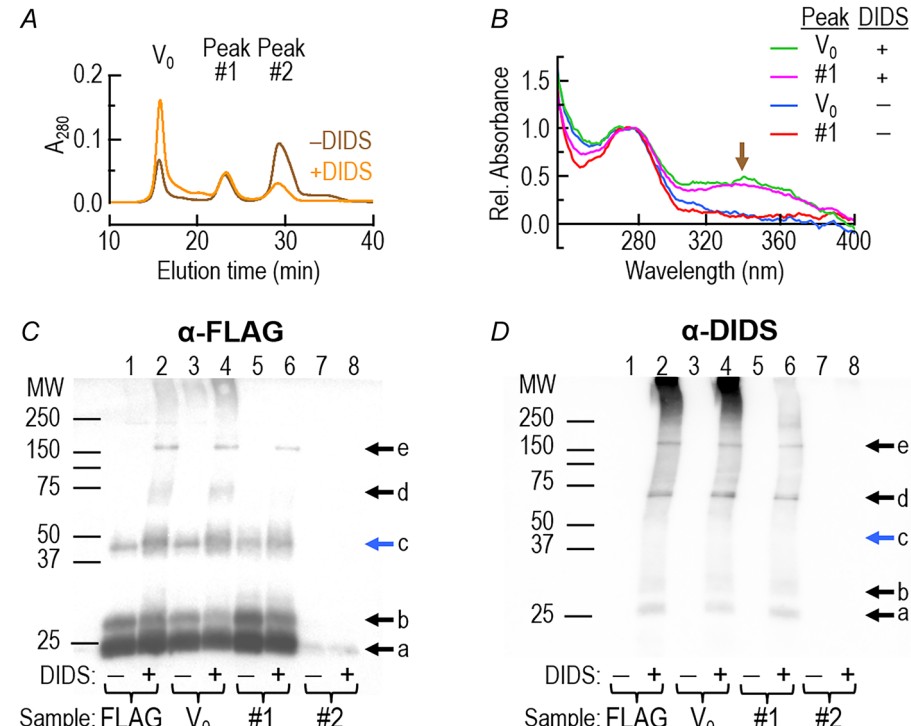

**Figure 10. Reaction of DIDS with N-terminally FLAG-tagged hAQP1 overexpressed in *Pichia pastoris***
A, Size-exclusion chromatography of solubilised proteins. We treated spheroplasts with DIDS (or no DIDS in parallel, sham experiments), solubilised with DMM, purified N-terminally FLAG-tagged hAQP1-WT using an anti-FLAG column, and then separated by FPLC on a Superdex-200 column, recording A$_{280}$ vs. time. The orange record represents a DIDS-treated sample and the brown record, a control sample (–DIDS) from the same preparation. The first peaks are the void volume (V$_0$), which consists of high molecular-weight (MW) proteins. Peak #1 consists of lower-MW proteins, and Peak #2, even smaller ones. We translated both records so that A$_{280}$ averaged zero between 10 and 13 min, and then we scaled the orange +DIDS record so that it has the same total area under the curve as Peak #1 in the –DIDS record. B, Absorbance spectra of material (±DIDS) from peaks V$_0$ and #1, from a preparation similar to that shown in panel A. We normalised all spectra to unity at 280 nm. The increased absorbance from ∼320 to ∼370 nm in the DIDS-treated samples presumably represents DIDS, which has an absorbance peak at ∼340 nm (Kodippili et al., 2009). C, Western blot of material from the same experiment as in panel B, probed with anti-FLAG: Lane 1. Material obtained from spheroplasts not treated with DIDS, and purified on an anti-FLAG column, as described for panel A, but not subjected to size-exclusion chromatography. ||| Lane 2. Same as Lane 1, but from DIDS-treated spheroplasts ||| Lane 3. Peak V$_0$ from spheroplasts –DIDS ||| Lane 4. Same as Lane 3, but from spheroplasts +DIDS ||| Lane 5. Peak #1 from spheroplasts –DIDS ||| Lane 6. Same as Lane 5, but from spheroplasts +DIDS ||| Lane 7. Peak #2 from spheroplasts –DIDS ||| Lane 8. Same as Lane 7, but from spheroplasts +DIDS. The presumed assignments are: 'a' band <25 kDa, unglycosylated hAQP1 monomer cleaved in the C-terminus; band 'b', 28 kDa, full-length unglycosylated or core-glycosylated monomer; band 'c', mature glycosylated monomer (blue arrow); band 'd' glycosylated dimer; band 'e', glycosylated tetramer. D, Western blot of material from the same experiment as depicted in panels B and C, probed with anti-DIDS. The lane assignments are the same as in panel C. The results in panels A through D are representative of two independent experiments. The absence of i in lanes 1, 3 and 5, especially where there is corresponding immunoreactivity for FLAG-tagged hAQP1-WT in panel C, demonstrates the specificity of the anti-DIDS antibody.

conclude that the monomeric pathway accounts for about half the total $(\Delta pH_S^*)_{CO_2}$ signal. Supporting evidence for this conclusion in the present study is that pCMBS reduces $(\Delta pH_S^*)_{CO_2}$ by somewhat more than half (Fig. 4A, second bar from left vs. leftmost bar), an effect abrogated by the C189S mutation (Fig. 4B, analogous bars). That some $CO_2$ may move through the monomeric pore is not unexpected, given the amphiphilic nature of $CO_2$. Moreover, the molecular dynamics simulations of Wang et al. (2007) suggest that some of the $CO_2$ permeability of hAQP1 is due to the 4 monomeric pores.

Is it possible that virtually all of the $CO_2$ that moves through hAQP1 does so by transiting the monomeric pores? Because treatment with $HgCl_2$ or pCMBS blocks only half of $P_f^*$, one might argue by analogy that this pCMBS might block only about half of the $CO_2$ traffic through the monomeric pores (i.e., the blocked and unblocked components of $CO_2$ traffic through the monomeric pore would sum to $\sim$100% of all $CO_2$ traffic through hAQP1). We believe that this is unlikely for two reasons: (1) $CO_2$ is a larger molecule (van der Waals volume $\cong$ 43 mL/mol) than either $NH_3$ ($\sim$37 mL/mol) or $H_2O$ ($\sim$31 mL/mol). (2) DIDS blocks a major component of $(\Delta pH_S^*)_{CO_2}$ and yet has no effect on either $(\Delta pH_S^*)_{NH_3}$ or $P_f^*$. The monomeric-pore-only hypothesis for $CO_2$ could be true only if DIDS blocks a major component of $CO_2$ movement through the monomeric pores and yet does not affect either $NH_3$ or $H_2O$ movement through the monomeric pores, which we believe to be unlikely.

Is it possible that pCMBS does not block all $CO_2$ traffic through the monomeric pore? Because (1) pCMBS+DIDS blocks all $CO_2$ traffic through hAQP1 and (2) DIDS has no effect on $NH_3$ or $H_2O$ traffic (which occurs exclusively via monomeric pores), the most straightforward explanation for our data is that pCMBS blocks nearly all of the $CO_2$ traffic that occurs through monomeric pores.

**$CO_2$ permeation through a parallel pathway.** Assuming that DIDS does not block the monomeric pore (see previous sentence), the present work provides the first physiological evidence for a second pathway for $CO_2$ through any AQP tetramer. This alternative pathway is apparently parallel to the four monomeric pores. The primary evidence is that (1) DIDS reduces $(\Delta pH_S^*)_{CO_2}$ by somewhat more than half in hAQP1-WT (Fig. 4A, third bar from the left vs. leftmost bar), (2) the C189S mutation does not affect the DIDS blockade (Fig. 4B, analogous bars), and (3) the combination of pCMBS and DIDS reduces $(\Delta pH_S^*)_{CO_2}$ to zero for oocytes expressing hAQP1-WT (Fig. 4A, rightmost vs. leftmost bars), but stillreduces it by somewhat more than half for oocytes expressing hAQP1-C189S (Fig. 4B, analogous bars).

**Molecular dynamics.** The simulations of Wang et al. (2007) suggest that a major component of the $CO_2$ flux

through hAQP1 occurs via the hydrophobic central pore that is largely devoid of $H_2O$ because it is lined by the sidechains of the following residues (from extracellular to intracellular sides): Val-50, Leu-54, and Leu-58 (all contributed by TM2), and Leu-174 and Leu-170 (from TM5). The mobility of a gas like $CO_2$ through such a near vacuum is $\sim$10$^4$ higher than in liquid water (see Boron, 2010; Rumble, 2022). Thus, even though central pores may comprise only a small fraction of total membrane surface area, it is possible that such pores—on the background of a membrane with a low intrinsic $CO_2$ permeability in the absence of AQP1 (Boron et al., 2011)—could make a significant contribution to overall $CO_2$ permeability. We propose that the central pore is the anatomic substrate of the DIDS-sensitive component of $CO_2$ permeability of hAQP1-WT.

### Effect of inhibitors

**pCMBS.** In Results, together with the presentation of Fig. 2D, we noted that the reaction of AQP1-C189 (R-SH) with $HgCl_2$ (as used by Preston et al., 1993) yields R-S-Hg-Cl as opposed to the reaction with pCMBS (Cl-Hg-R$'$), which yields R-S-Hg-R$'$. Thus, $HgCl_2$ treatment yields a product that has steric effects, whereas pCMBS treatment yields a product with even greater steric effects and the electrostatic effects of the sulfonate group. Although the two agents could very well have different inhibitory profiles within the monomeric pore, it is interesting that we found that pCMBS reduces $P_f^*$ to about the same extent as previously observed with $HgCl_2$ by Preston et al. (1993).

**DIDS.** With its two disulfonate groups, the large DIDS molecule (i.e., DIDS $^=$) is virtually membrane impermeant. In the case of its interactions with AE1 (Cabantchik & Rothstein, 1972) and NBCe1 (Lu & Boron, 2007), the first interaction of DIDS with a target is a reversible ionic interaction (e.g., with a cluster of protonated lysine groups). The higher the affinity of an ionic binding site for the drug, the greater the chance of one of the two –SCN groups of DIDS to be near a lysine residue as it temporarily deprotonates, permitting the covalent reaction as noted in the presentation of Fig. 2E. Once one –SCN group reacts with one target lysine residue, the odds of the second – SCN undergoing a similar reaction with a nearby lysine—the crosslinking reaction—rise enormously in what is known as an avidity effect. In the cases of AE1 and NBCe1, both the reversible ionic interactions and the irreversible covalent reactions block transport.

Because its blockade is irreversible with both hAQP1-WT (Fig. 9A) and hAQP1-FLAG (Fig. 9B), the DIDS interaction with hAQP1 expressed in oocytes

has probably already advanced to the formation of at least one covalent bond. We propose that DIDS, via a covalent reaction, fully blocks the parallel pathway of nearly all otherwise-functional hAQP1 tetramers on the oocyte surface. If this parallel pathway is the central pore, then the DIDS could either (1) covalently interact with a single monomer, ±crosslinking, in such a way as to obstruct the central pore but not the monomeric pore or (2) covalently crosslink two monomers to obstruct the central pore. We recognise that DIDS could, in principle, form multiple types of single-covalent or crosslinking-covalent bonds within a hAQP1 tetramer, but that not all may produce the physiological blockade of $CO_2$ conductance that we observe in the present oocyte studies.

In our work with DIDS-treated *Pichia* cells, western blots probed with a DIDS antibody (Fig. 10*D*) reveal presumed dimers and tetramers, but virtually no glycosylated monomers (compare Fig. 10*C* vs. *D*, blue arrows). These results imply that, at least in *Pichia*, once one end of a DIDS molecule reacts with one lysine residue on a hAQP1 monomer at the cell surface, the probability is extremely high (avidity effect) that the opposite end cross-links to a lysine residue on a different monomer.

The likely targets of DIDS on hAQP1 are two extracellular-facing lysine residues, K36 (on loop A between TM1 and TM2) and K51 (near the beginning of TM2, which lines the extracellular end of the central pore). Because a tetramer has $4 \times 2$ such lysines, seven unique types of DIDS crosslinking are possible: within a monomer (1 type), between adjacent monomers (3), and between monomers on opposite sides of the central pore (3). Given the length of the DIDS molecule (see Fig. 2*E*), Dr. Ardi Vahidi-Faridi (personal communication) identifies the three most likely crosslinks as K36-K51 within the same monomer, K36-K51 of adjacent monomers, and K51-K51 of adjacent monomers.

**Summing the effects of pCMBS and DIDS.** The mathematical simulations of Somersalo et al. (2012) suggest that both $(\Delta pH_S)_{CO2}$ and $(dpH_i/dt)_{Max}$ have similar sigmoidal dependencies on the $\log_{10}$ of the membrane permeability to $CO_2$ ($P_{M,CO2}$). Our physiological data must fall somewhere in the range where $(\Delta pH_S)_{CO2}$ is predicted to rise approximately linearly with $\log_{10}(\Delta pH_S)_{CO2}$ (see their Fig. 7*B*). If we assume that a $H_2O$-injected oocyte falls at the lower end of this linear range (their dark-grey point representing $P_{M,CO2} = [34.2 \text{ cm s}^{-1}]/[5 \times 10^4]$), then a doubling of $(\Delta pH_S)_{CO2}$—representing the contribution of hAQP1—would require that we increase $P_{M,CO2}$ by ~5-fold (their magenta point for $P_{M,CO2} = [34.2 \text{ cm s}^{-1}]/[1 \times 10^4]$). Starting from this relatively large $(\Delta pH_S)_{CO2}$, reducing the $\Delta pH_S$ by half—representing the inhibition by DIDS or pCMBS—would require that we

decrease $P_{M,CO2}$ by nearly 60% (their gold point for $P_{M,CO2} = [34.2 \text{ cm s}^{-1}]/[2.5 \times 10^4]$), which represents a nearly 75% reduction of the hAQP1-dependent component of $P_{M,CO2}$. If we were to start our analysis further down the sigmoid curve, the predicted inhibition of $P_{AQP,CO2}$, by each drug would be somewhat less than 75%, whereas if we had started further up the curve, the predicted inhibition of the Somersalo model would be much higher than ~75%.

If pCMBS and DIDS each decrease $(\Delta pH_S^*)_{CO2}$ by ~50%—and if we initiate our analysis at the dark-grey point (Fig. 7*B* of Somersalo et al., 2012)—then each drug must produce a far greater fractional decrease in $P_{AQP,CO2}$. Following the logic of the mathematical simulations, the blockade of the monomeric pore by pCMBS should be accompanied by some degree of blockade of the parallel $CO_2$ pathway (e.g., central pore). However, the converse is apparently not true. That is, the blockade of the parallel pathway by DIDS seems not to produce significant effects on the four monomeric pores inasmuch as DIDS has no effect on either $(\Delta pH_S^*)_{NH3}$ (Fig. 7) or $P_f^*$ (Fig. 8). Such an overlapping effect of pCMBS on the monomeric and parallel pathways is not unreasonable inasmuch as the target of pCMBS (i.e., C189) is near the extracellular-facing NPA motif and the selectivity filter of the monomeric pore, both of which are in close proximity to TM2, which lines the extracellular half of the central pore. Thus, it is possible that derivatization of C189 by pCMBS could lead to shifting or twisting of TM2, thereby reducing gas permeability through the central pore. We propose that (1) pCMBS not only blocks all $CO_2$ and $NH_3$ traffic through the monomeric pore but also part of the $CO_2$ traffic through the parallel pathway, and (2) DIDS blocks (most) $CO_2$ traffic through the parallel pathway but no traffic through the monomeric pores. These hypotheses are consistent with the data of the present paper. Given the approximately log-linear relationship between $P_{M,CO2}$ and $(\Delta pH_S)_{CO2}$—and assuming that pCMBS and DIDS each reduce $P_{AQP,CO2}$ by ~75% (see previous paragraph)—we propose that ~25% of the hAQP1-dependent $CO_2$ traffic occurs through the four monomeric pores (fully blocked by pCMBS but unaffected by DIDS) whereas ~75% of this $CO_2$ traffic occurs through the single central pore (partially blocked by pCMBS but fully blocked by DIDS).

## Molecular basis of gas selectivity

An important unresolved issue is the molecular basis for gas selectivity by AQPs (Geyer et al., 2013; Musa-Aziz, Chen et al., 2009).

**$CO_2$.** The most straightforward interpretation of the data in the present study is that the amphiphilic $CO_2$ molecule transits hAQP1 via 2 pathways: the four hydrophilic

monomeric pores (presumably a relatively small fraction, $\sim$25%) and a parallel pathway (e.g., hydrophobic central pore).

**NH₃.** The most straightforward interpretation of the present data is that all $NH_3$ transits the four hydrophilic monomeric pores of hAQP1; that is, none moves via the parallel pathway (e.g., hydrophobic central pore). Considering the broader family of AQPs, we predict that the hydrophobic parallel pathways (e.g., central pores) are never important routes of $NH_3$ conductance, but rather that the characteristics of the monomeric pores determine whether a particular AQP has a relatively high permeability to $NH_3$ (AQPs 3, 6, 7, 8, 9), or a relatively low $NH_3$ permeability (AQPs 2, 4, 5).

We suggest that, for the AQPs known to be permeable to $CO_2$ (AQPs 0, 4, 5, 6, 9), $CO_2$ transits some combination of monomeric pores and a parallel, more hydrophobic pathway. For the AQPs with limited $CO_2$ conductance (AQPs 2, 3, 7, 8), both the monomeric pores and the parallel pathways presumably have less favourable physico-chemical characteristics. Because more-hydrophobic molecules like $O_2$ and $N_2$ are less likely to transit through the monomeric pores, their permeabilities through various AQPs likely depend mainly on the physico-chemical characteristics of the parallel, hydrophobic pathway. Thus, the gas-selectivity profile of each AQP likely depends on the physico-chemical nature of each gas and the unique set of physico-chemical properties of the (at least 2) potential pathways through the tetramer.

## Concluding remarks

Our work establishes the proof of principle for blocking two pathways for $CO_2$ permeation through hAQP1. We suggest that the pCMBS, even though it targets the monomeric pore and reduces $CO_2$ permeation via this route, also reduces $CO_2$ permeability through the parallel pathway. We hypothesise that the DIDS almost exclusively inhibits the parallel pathway. If it were possible to eliminate the monomeric pathway and make judicious mutations to the hydrophobic alternate pathway, one might be able to design channels with exquisite selectivity amongst gases. Such designer channels could improve industrial gas handling on a micro- or nanoscale, provide a simple approach for venting metabolically produced $CO_2$ in diving or space flight, provide control of $CO_2$ and $N_2$ conduction in agriculture, and enable precise $O_2$ control for medical therapies, synthetic biology and life-support systems in underwater or space environments.

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

## Additional information

### Data availability statement

All raw data are deposited into the NIH-supported Zenodo data repository in common and open formats and are accessible via the persistent identifier DOI: 10.5281/zenodo.16966460.

### Competing interests

The authors declare that they have no competing interests.

### Author contributions

R.M.-A. contributed to the conception and design of the research; performed oocyte experiments; analyzed and

interpreted the oocyte data; prepared figures; drafted and edited the manuscript. R.R.G. contributed to the conception and design of the research related to oocyte $P_f$ experiments as well as Pichia pastoris research; performed all the Pichia pastoris experiments; analyzed and interpreted all results collected with Pichia pastoris; prepared figures; and drafted the manuscript. S-K.L. validated the cDNA clones for hAQP1 used in the experiments and contributed to the interpretation of pHS experiments. F.J.M. contributed to the design of the research; prepared the figures; analyzed the statistics and interpreted results; drafted and edited the manuscript. W.F.B contributed to the conception and design of the research, and also edited the manuscript. All authors approved the final version of the manuscript and all qualify for authorship, and all those who qualify for authorship are listed.

## Funding

This work was supported by Fundação de Amparo à Pesquisa do Estado de São Paulo, Office of Naval Research, National Institute of Neurological Disorders and Stroke, National Institute of Diabetes and Digestive and Kidney Diseases, National Heart, Lung, and Blood Institute, National Institute of General Medical Sciences and Multidisciplinary University Research Initiative.

## Acknowledgements

The Article Processing Charge for the publication of this research was funded by the Coordenação de Aperfeiçoamento de Pessoal de Nível Superior - Brasil (CAPES) (ROR identifier: 00x0ma614). We also thank Duncan Wong for helpful discussions and computer support; Dale Huffman for engineering assistance and helpful discussions. We thank Steven Torontali for designing and fabricating the oocyte chamber; Gerald Babcock for technical assistance and laboratory management; Morley Schwebel; Charleen Bertolini, Rosalyn Forster, and Lesa Goodman for administrative support. We thank Dr. Ardi Vahedi-Faridi for helpful discussions on the potential targets of DIDS on hAQP1, as well as Prof. Leslie Poole and Prof. S. Bruce King for helpful discussions on the interactions of mercurial compounds with –SH groups. W. F. Boron gratefully acknowledges the support of the Myers/Scarpa endowed chair. We thank Dr. Pan Zhao for providing the structural formulas for pCMBS and DIDS. R. Musa-Aziz gratefully acknowledges Prof. Mark D. Parker (University at Buffalo: State University of New York) for encouragement and helpful discussions, and gives special thanks in memory of Dr. Gerhard Malnic (University of São Paulo, SP, BR) for his unwavering support and advice.

[Correction added on 11 December 2025 after first online publication: The Acknowledgements section has been added.]

## Keywords

CO$_2$ permeability, osmotic water permeability, Surface pH, Xenopus laevis oocytes

## Supporting information

Additional supporting information can be found online in the Supporting Information section at the end of the HTML view of the article. Supporting information files available:

**Peer Review History**

