## [Peer Review History · The Journal of Physiology]

Mechanism of CO₂ and NH₃ Transport through Human Aquaporin 1: Evidence for Parallel CO₂ Pathways

Raif Musa-Aziz, Robert Ryan Geyer, Seong-Ki Lee, Fraser J Moss, and Walter F Boron
DOI: 10.1113/JP289556

Corresponding author(s): Raif Musa-Aziz (raifaziz@icb.usp.br)

The following individual(s) involved in review of this submission have agreed to reveal their identity: Gordon J Cooper (Referee #2)

Review Timeline:

Submission Date:	23-Jun-2025
Editorial Decision:	11-Aug-2025
Revision Received:	25-Sep-2025
Editorial Decision:	04-Nov-2025
Revision Received:	05-Nov-2025
Accepted:	10-Nov-2025

Senior Editor: Kim Barrett

Reviewing Editor: Peking Fong

Transaction Report:

Dear Dr Musa-Aziz,

Re: JP-RP-2025-289556 "Mechanism of CO₂ and NH₃ Transport through Human Aquaporin 1: Evidence for Parallel CO₂ Pathways" by Raif Musa-Aziz, Robert Ryan Geyer, Seong-Ki Lee, Fraser J Moss, and Walter F Boron

Thank you for submitting your manuscript to The Journal of Physiology. It has been assessed by a Reviewing Editor and by 2 expert referees and we are pleased to tell you that it is potentially acceptable for publication following satisfactory major revision.

REVISION CHECKLIST:

Please upload two versions of your manuscript text: one with all relevant changes highlighted and one clean version with no

changes tracked. The manuscript file should include all tables and figure legends, but each figure/graph should be uploaded as separate, high-resolution files.

We look forward to receiving your revised submission.

Yours sincerely,

Kim Barrett
Senior Editor
The Journal of Physiology

REQUIRED ITEMS

- 1) - Author photo and profile. First or joint first authors are asked to provide a short biography (no more than 100 words for one author or 150 words in total for joint first authors) and a portrait photograph. These should be uploaded and clearly labelled together in a Word document with the revised version of the manuscript. See Information for Authors for further details.
- 2) - You must upload original, uncropped western blot/gel images (including controls) if they are not included in the manuscript. This is to confirm that no inappropriate, unethical or misleading image manipulation has occurred. These should be uploaded as 'Supporting information for review process only'. Please label/highlight the original gels so that we can clearly see which sections/lanes have been used in the manuscript figures. For more information, see: <https://physoc.onlinelibrary.wiley.com/hub/journal-policies#imagmanip>.
- 3) - Please ensure that any tables are editable and in Word format, and wherever possible, embedded in the article file itself.
- 4) - Please ensure that the Article File you upload is a Word file.
- 5) - Please include an Abstract Figure file, as well as the Figure Legend text within the main article file. The Abstract Figure is a piece of artwork designed to give readers an immediate understanding of the research and should summarise the main conclusions. If possible, the image should be easily 'readable' from left to right or top to bottom. It should show the physiological relevance of the manuscript so readers can assess the importance and content of its findings. Abstract Figures should not merely recapitulate other figures in the manuscript. Please try to keep the diagram as simple as possible and without superfluous information that may distract from the main conclusion(s). Abstract Figures must be provided by authors no later than the revised manuscript stage and should be uploaded as a separate file during online submission labelled as File Type 'Abstract Figure'. Please also ensure that you include the figure legend in the main article file. All Abstract Figures should be created using BioRender. Authors should use The Journal's premium BioRender account to export high-resolution images. Details on how to use and access the premium account are included as part of this email.

EDITOR COMMENTS

Reviewing Editor:

This is a well-performed study investigating the routes of carbon dioxide and ammonia transit through aquaporin 1. There is an emphasis on discerning whether the central pore of aquaporin 1 permeates carbon dioxide by combining functional and biochemical assays in what essentially amounts to a subtractive analysis.

The Referees identify several critical assumptions requiring further justification. Specifically, these focus on the effects of pCMBS, as voiced strongly by Referee 1, and hinted at by Referee 2 (next to last point in Major Comments).

The Referees concur on the need for additional data to assess protein expression of wild type and C189S aquaporin 1 in a

quantitative and statistically robust manner. For thoroughness, showing the blot across a broader molecular weight range might be useful for assessing multimeric assemblies resulting from wild type and C189S aquaporin 1 overexpression in *Xenopus* oocytes, as done in figure 10 for the *P. pastoris*-derived preparation.

The study might be improved by using the next generation aquaporin 1 modulators mentioned by Referee 2 in similar analyses.

For the most part, the data are presented in a statistically rigorous fashion, so much so that several lapses (caught by Referee 2; last five points under "Major comments") bear mention.

The writing is clear overall, but nonetheless sections do require correction. Some idiosyncratic constructions merit reconsideration and several sections within the Methods can be written more economically without sacrificing content. For specifics, please refer to comments offered by Referee 2 under subheading "Minor comments".

In addition, I can offer a few thoughts pertaining to Figure 10C and 10D for consideration. In panel C, the signal detected for the glycosylated monomer (c-band) by the anti-FLAG antibody can be appreciated as lower than that of the unglycosylated, uncleaved (b-band) and unglycosylated, cleaved (a-band) monomers. However, panel D shows the anti-DIDS antibody nonetheless detects signal for the latter two, whereas there is a conspicuous, almost complete absence of c-band. This is surprising; can any signal be revealed by overexposure? Here is an instance where blot quantification (and clearer demonstration of reproducibility) would be highly informative. Perhaps IP with anti-FLAG and then probing with anti-DIDS might be cleaner.

In addition, the legend to figure 2C appears to contain a section at the end that likely should appear in the description of figure 2A. Please correct this.

REFeree COMMENTS

Referee #1:

The manuscript "Mechanism of CO₂ and NH₃ transport through human aquaporin 1: evidence for parallel CO₂ pathways" by Musa-Aziz and colleagues uses physiologic approaches to assess the pathway(s) through which CO₂ and NH₃ are transported by hAQP1. The authors are experts in this field and in the techniques which are used in the manuscript.

Major comments:

1. The conclusion that hAQP1 transports CO₂ via both the monomeric pore and through a separate pathway is predicated upon the unstated but explicit assumption that pCMBS has a 100% inhibitory effect on human AQP1 monomeric transport. While this appears to be reasonable for NH₃ transport, slight differences in the molecular mechanisms of CO₂ and NH₃ transport might result in different inhibitory efficacy of pCMBS for these two molecules. If this assumption is not correct, the entire conclusion that a separate pathway for human AQP1 CO₂ transport is present is lost. Studies using mutated hAQP1 in which key residues necessary for monomeric pore transport have been mutated may be needed to conclude that the monomeric pore does not mediate 100% of human AQP1 CO₂ transport.

Minor comments:

1. The finding that pCMBS blocks only 50% of AQP1 CO₂ transport has been shown previously. This should be more explicitly stated in the manuscript.
2. The anti-DIDS antibodies used in the studies do not appear to be affinity-purified. Is this correct? Can evidence of their specificity, using DIDS as a blocking antigen, be shown? Were comparable results obtained with antibodies from both animals?
3. How many replicates of the data in Figure 2C were performed? Given that similar levels of expression of the two AQP1 proteins, wild-type and mutated, are essential for interpretation of the studies comparing transport of the 2 AQP1 proteins in this manuscript, can a summary figure showing the replicated data be added?
4. What is the evidence that the effect of pCMBS is irreversible?
5. Can the prior molecular dynamic simulation studies showing NH₃ movement through the monomeric AQP1 pore be discussed more explicitly?

Referee #2:

The current study provides a series of carefully performed and details experiments investigating the mechanisms by which

water, ammonia and carbon dioxide are transported through AQP1. Based on selective inhibitors the data clearly provides evidence for two distinct pathways through AQP1 which possess different permeability properties. The monomeric AQP1 pores show permeability to all three substances while a DIDS selective pathway, likely through the central pore of the AQP1 tetramer allows passage of carbon dioxide but not water or ammonia. This work reinforces previous studies and inclusion of the C189S mutant data builds on the previous studies. The use of the anti-DIDS antibody provides some exciting insights into the mechanism of action of this drug, which appears to exert an action by crosslinking between the AQP1 monomers that form the tetramer.

General thoughts:

There are now several AQP1 modulators available which are specific for the AQP1 monomeric pore or block the cationic conductance. Have the authors investigated the actions of any of these compounds which would help support the conclusions of the current study.

While the authors have provided evidence that DIDS is crosslinking the AQP1 monomeric units the paper would be enhanced if they were able to provide a molecular basis for this action. It is well established that DIDS interacts with Lysine residues and a KXXXK motif has been proposed as the basis of DIDS action in anion exchangers. Although AQP1 does not have such a motif, the K51 residue lines the tetrameric pore. Are the authors able to speculate on whether this would be a viable target for DIDS.

Paper review. Major comments:

The introduction provides a detailed overview of the literature linked to the transport of gases by Aquaporin. There is limited mention of the role of AQP1 as an ion channel and the potential roles of the monomeric and/or central pores in this action. There is evidence to suggest that in these cases the central pore may be permeable to a substance (as mentioned in line 157) so it would be beneficial to expand the introduction to incorporate this information and put it in context with the current study. Additionally the Al-Samir (2025) study incorporates inhibitors which act on the ion conductance pathway, and it would be useful to address this in light of your current findings.

Overall the methods section feels very long for a paper of this nature. Although it is extremely detailed, in many places it is describing well established processes which have been described in depth previously. I feel the manuscript would benefit from the methods section being more concise.

Line 464 - Pf assays. Can the authors confirm in this section how long the oocytes were in the hypotonic solution - is it 60s as suggested in the earlier methods. Based on previous studies using a similar construct, injection of oocytes with 25ng of cRNA is likely to create high expression of AQP1 protein with potential damage to the oocytes. Were the authors happy of the membrane integrity at the end of the first hypotonic exposure and how did they verify this.

Line 583 and Figure 2C. The authors have presented evidence that AQP1 and its C189S mutant are expressed at similar levels in oocytes. Can they please confirm if this was 3 preparations from different animals. Have the blots been quantified in any way to confirm similar levels of protein expression.

Line 592 and Figure 3. The authors indicate the pHS deflection is much smaller in control oocytes compared to oocytes expressing AQP1 or its C189s mutant. Have you confirmed this statistically? Given the other sections of the manuscript rely on the difference between the two values to give the channel dependent component of CO₂ transport it is crucial to state the significance.

Line 660 and Figure 6 - See comments above for line 592 and Figure 3, the same idea applies.

Line 606-611 Could the authors include statistics to confirm that pCMBS has an effect on AQP1 but not the C189S mutant.

Line 613 - Can the authors include statistics to indicate the (Δ pHs*)CO₂ for AQP1 and C189S are very similar and not different from each other.

Minor comments:

Line 39 - Please check this doesn't make sense

Line 46 - after "by" please add a value

Line 89 - "of course" - not necessary

Page 7 - Comment 1 - not needed

Line 185 - Please add details of Rabbit husbandry

Line 214 - Should "Solutions 4 and 5" be Solution 5 only. The description here is linked to the preparation of the 0.5mM NH₃/NH₄ solution

Line 564 The mechanism of the pH changes has been described in detail previously and this explanation could be shortened.

Line 602 - The Preston et al, 1993 reference evaluates the effect of HgCl₂ on the C189S and does not address pCMBS. A different reference may be more appropriate here.

Line 671 - You indicate 11 oocytes in the text and 12 in Figure 6B - please check and confirm correct value

Line 688 - The Macey (1984) study investigated the effect of mecurials on RBCs and not oocytes. Please consider if this reference is appropriate to be used here.

END OF COMMENTS

Dear Dr Musa-Aziz,

Re: JP-RP-2025-289556 "Mechanism of CO₂ and NH₃ Transport through Human Aquaporin 1: Evidence for Parallel CO₂ Pathways" by Raif Musa-Aziz, Robert Ryan Geyer, Seong-Ki Lee, Fraser J Moss, and Walter F Boron

Thank you for submitting your manuscript to The Journal of Physiology. It has been assessed by a Reviewing Editor and by 2 expert referees and we are pleased to tell you that it is potentially acceptable for publication following satisfactory major revision.

Your revised manuscript should be submitted online using the link in your Author Tasks: <https://jp.msubmit.net/cgi-bin/main.plex?el=A3JS3HWQ7A2Vje3F1A9ftdApyMXYmaP5Q0mZ5iaARxQZ>.

This link is accessible via your account as Corresponding Author; it is not available to your co-authors. If this presents a problem, please contact journal staff (jp@physoc.org). Image files from the previous version are retained on the system. Please ensure you replace or remove any files that are being revised.

LANGUAGE EDITING AND SUPPORT FOR PUBLICATION: If you would like help with English language editing, or other article preparation support, Wiley Editing Services offers expert help, including English Language Editing, as well as translation, manuscript formatting, and figure formatting at www.wileyauthors.com/eo/preparation. You can also find resources for Preparing Your Article for general guidance about writing and preparing your manuscript at www.wileyauthors.com/eo/prepresources.

REVISION CHECKLIST:

We look forward to receiving your revised submission.

Yours sincerely,

Kim Barrett

Senior Editor

The Journal of Physiology

REQUIRED ITEMS

1) - Author photo and profile. First or joint first authors are asked to provide a short biography (no more than 100 words for one author or 150 words in total for joint first authors) and a portrait photograph. These should be uploaded and clearly labelled together in a Word document with the revised version of the manuscript. See Information for Authors for further details.

2) - You must upload original, uncropped western blot/gel images (including controls) if they are not included in the manuscript. This is to confirm that no inappropriate, unethical or misleading image manipulation has occurred. These should be uploaded as 'Supporting information for review process only'. Please label/highlight the original gels so that we can clearly see which sections/lanes have been used in the manuscript figures. For more information, see: <https://physoc.onlinelibrary.wiley.com/hub/journal-policies#imagmanip>.

Response: The blots presented in Figure 10C & D are already uncropped. We will upload the uncropped and annotated blot image for the part used in Figure 2C.

Action: The uploaded file “Uncropped_WB_Fig2C.pptx” is designated “Supporting information for review process only”.

3) - Please ensure that any tables are editable and in Word format, and wherever possible, embedded in the article file itself.

4) - Please ensure that the Article File you upload is a Word file.

5) - Please include an Abstract Figure file, as well as the Figure Legend text within the main article file. The Abstract Figure is a piece of artwork designed to give readers an immediate understanding of the research and should summarise the main conclusions. If possible, the image should be easily 'readable' from left to right or top to bottom. It should show the physiological relevance of the manuscript so readers can assess the importance and content of its findings. Abstract Figures should not merely recapitulate other figures in the manuscript. Please try to keep the diagram as simple as possible and without superfluous information that may distract from the main conclusion(s). Abstract Figures must be provided by authors no later than the revised manuscript stage and should be uploaded as a separate file during online submission labelled as File Type 'Abstract Figure'. Please also ensure that you include the figure legend in the main article file. All Abstract Figures should be created using BioRender. Authors should use The Journal's premium BioRender account to export high-resolution images. Details on how to use and access the premium account are included as part of this email.

EDITOR COMMENTS

Reviewing Editor:

This is a well-performed study investigating the routes of carbon dioxide and ammonia transit through aquaporin 1. There is an emphasis on discerning whether the central pore of aquaporin 1 permeates carbon dioxide by combining functional and biochemical assays in what essentially amounts to a subtractive analysis.

The Referees identify several critical assumptions requiring further justification. Specifically, these focus on the effects of pCMBS, as voiced strongly by Referee 1, and hinted at by Referee 2 (next to last point in Major Comments).

Response: Thank you, Dr. Barrett, for your thoughtful and constructive feedback. We have carefully addressed each major and minor point of each reviewer and believe these revisions have strengthened the manuscript.

Action: We have revised the manuscript accordingly and prepared a detailed point-by-point "Response to Referees" document.

The Referees concur on the need for additional data to assess protein expression of wild type and C189S aquaporin 1 in a quantitative and statistically robust manner. For thoroughness, showing the blot across a broader molecular weight range might be useful for assessing multimeric assemblies resulting from wild type and C189S aquaporin 1 overexpression in *Xenopus* oocytes, as done in figure 10 for the *P. pastoris*-derived preparation.

Response: Agree. Note that Figure 10 reveals dimers and tetramers only in DIDS-treated samples, which is why they are not visible here.

Action: See response to Referee #1/Minor comment #3, which includes the stats and a new version of the photo with MW extended to 100 kDa.

The study might be improved by using the next generation aquaporin 1 modulators mentioned by Referee 2 in similar analyses.

Response: As noted in the response to Referee #2/General thoughts, it is not clear what agents he/she is referring to. It is well known that the AQPs have been notoriously difficult drug targets (see Verkman ref.). Moreover, Hg agents like pCMBS (which simply adds an Hg atom to Cys) act by a well-documented mechanism.

For the most part, the data are presented in a statistically rigorous fashion, so much so that several lapses (caught by Referee 2; last five points under "Major comments") bear mention.

Response: These were worthwhile exercises: The first dealt with Fig 2C (see above). The next two (are the AQP1 signals > H₂O controls/background?) were fairly trivial, inasmuch as the background-subtracted data that we presented (i.e., $(\Delta pH_S^*)_{CO_2}$, and $(\Delta pH_S^*)_{NH_3}$) both were very different from zero. The last 2 asked us to compare control data between panels (the bars were obviously of similar height).

Action: We acceded to all 5 requests as detailed in the responses to Referee #2

The writing is clear overall, but nonetheless sections do require correction. Some idiosyncratic constructions merit reconsideration and several sections within the Methods can be written more economically without sacrificing content. For specifics, please refer to comments offered by Referee 2 under subheading "Minor comments".

Response: The Methods are lengthy because the protocol was so complex and we realized that previous papers did not provide a comprehensive outline of our approaches.

Action: We followed the Referee's suggestions for editing (including shortening of text), as described below.

In addition, I can offer a few thoughts pertaining to Figure 10C and 10D for consideration. In panel C, the signal detected for the glycosylated monomer (c-band) by the anti-FLAG antibody can be appreciated as lower than that of the unglycosylated, uncleaved (b-band) and unglycosylated, cleaved (a-band) monomers. However, panel D shows the anti-DIDS antibody nonetheless detects signal for the latter two, whereas there is a conspicuous, almost complete absence of c-band. This is surprising; can any signal be revealed by overexposure? Here is an instance where blot quantification (and clearer demonstration of reproducibility) would be highly informative. Perhaps IP with anti-FLAG and then probing with anti-DIDS might be cleaner.

Response: Thank you for the question. In retrospect, we realize that we could have written these passages more clearly. Two key points: (a) the DIDS is almost exclusively confined to the outside of the cells and (b) the less-than-mature-glycosylated AQP1 tetramers have a low probability of trafficking to the plasma membrane. Thus, bands 'a' and 'b' were poorly (if at all) accessible to extracellular DIDS.

With α FLAG in Fig 10C, the intensities of the monomers decrease in the order $a \gg \gg b \gg c$. With the α DIDS in Fig 10D, the monomers 'a' and 'b' are now much fainter (still, $a \gg b$), and thus we are not surprised that 'c' is virtually invisible. The results for 'a' and 'b' in Fig 10C vs. 10D are reasonable because DIDS is membrane impermeant and unlikely to label the intracellular 'a' and 'b', which should not traffic significantly to the plasma membrane.

A question that we had not addressed in the original ms was why, in the α FLAG blot in Fig 10C, should band "c" be more intense in the presence of DIDS. We do not know why ... perhaps the 1-hour DIDS treatment, which leads to crosslinking of AQP1 monomers (and interactions with other membrane proteins, too), leads to more mature glycosylation and trafficking of AQP1 to the plasma membrane. This increased signal in Fig 10C could then represent fully glycosylated AQP1 tetramers that reached the plasma membrane but did not yet react with DIDS before the end of the treatment. During preparation for the WT, these tetramers presumably dissociated to monomers that we see in Fig 10C (with α FLAG) but not in Fig 10D (with α DIDS).

One might worry that, in Fig 10D, bands 'a' and 'b' are so intense that they exhausted the ECL reaction and thus appear white. However, as noted above, we would not expect this cytosolic protein to be labeled by DIDS in the first place. Moreover, band 'a' is not white in Fig 10D, and the missing 'c' (at higher magnification) is in the middle of a sea of faint gray.

In the meantime, in a major effort that began with gallons of bovine blood, we believe that we (Ardi Vahedi-Faridi and Andreas Engel) have identified the target of DIDS by a radiolytic cleavage approach, and now have a high-resolution single-particle structure of bovine AQP1 with DIDS attached near the target lysine. We are working to assemble the ms that describes this work.

Action: We have revised the presentation in Results (see lines #753–802) and Discussion (see lines #918-931) to clarify the above points. In particular, we emphasize that DIDS is impermeant and bivalent and that bands a and b are very likely to be retained intracellularly. In addition, because the ms relies heavily on the two inhibitors, because the Referees have raised inhibitor-related questions, and because the chemistries of $HgCl_2$ (used, for example, by Agre) and the

much milder pCMBS are different (i.e., HgCl₂ leaves R-S-Hg-Cl whereas pCMBS leaves R-S-Hg-R', where R' is benzene sulfonate) we have added the structures as panels D and E of Fig 2. We have also added a paragraph (see lines #622-631) near the outset of Results to explain the above.

In addition, the legend to figure 2C appears to contain a section at the end that likely should appear in the description of figure 2A. Please correct this.

Response: Thank you.

Action: We moved the definitions to legend of panel A.

REFEREE COMMENTS

Referee #1:

The manuscript "Mechanism of CO₂ and NH₃ transport through human aquaporin 1: evidence for parallel CO₂ pathways" by Musa-Aziz and colleagues uses physiologic approaches to assess the pathway(s) through which CO₂ and NH₃ are transported by hAQP1. The authors are experts in this field and in the techniques which are used in the manuscript.

Response: Thank you! We very much appreciate your very positive comments.

Major comments:

Response: This first comment below (does all the CO₂ go through the monomeric pore?) led to editing of our presentation of this material in the Discussion.

1. The conclusion that hAQP1 transports CO₂ via both the monomeric pore and through a separate pathway is predicated upon the unstated but explicit assumption that pCMBS has a 100% inhibitory effect on human AQP1 monomeric transport. *[comment continues below]*

Response: We did not intend to make implicit assumptions. We have 1 key theoretical point (the electronic orbitals of NH₃ and H₂O are very similar) and 2 key experimental observations: (a) pCMBS blocks 100% of AQP1-dependent (ΔpH_S^*)_{NH₃}—i.e., eliminates NH₃ permeability through hAQP1—as demonstrated for the first time in the present paper. And (b) DIDS does not affect either (ΔpH_S^*)_{NH₃} or P_f^* .

Action: We have revised the Discussion (see lines #812–818) to state explicitly the 3 lines of evidence that NH₃ moves almost exclusively via the monomeric pore. In addition, we have added a paragraph in the Discussion (see lines #890–897) in which we consider the chemistry of the HgCl₂ vs. pCMBS reactions with C189.

[continuation of Referee's comment ...] While this appears to be reasonable for NH₃ transport, slight differences in the molecular mechanisms of CO₂ and NH₃ transport might result in different inhibitory efficacy of pCMBS for these two molecules.

Response: At the outset of the Discussion, we were remiss in not addressing the dichotomy that although Hg derivatization of C189 (near the selectivity filter of hAQP1) on the one hand reduces (ΔpH_S^*)_{NH₃} to zero but on the other only reduces P_f^* by about half. The latter is a confirmation of a well-established observation of others (see line #742). Thus, the system is complex. We agree that the chemistry governing the movements of particles through the monomeric pore (MP) of AQP1—and therefore the effects of Hg derivatization of C189—is likely to be different for H₂O, NH₃, and CO₂ (or O₂).

Action: We have revised the Discussion by introducing a new paragraph (see lines #819–825) that makes the above points.

[continuation of Referee's comment ...] If this assumption is not correct, the entire conclusion that a separate pathway for human AQP1 CO₂ transport is present is lost.

Response: As noted in the previous/previous Response, the conclusion that virtually all NH₃ traffic through hAQP1 is through the monomeric pore is supported by our DIDS data. That is, DIDS does not affect either NH₃ or H₂O permeability, and thus these hydrophilic molecules move exclusively via the hydrophilic monomeric pores and not the DIDS-sensitive alternate pathway.

Similarly to the conclusions regarding NH₃ and H₂O, the conclusion that some CO₂ moves through hAQP1 via a pathway other than the monomeric pore requires consideration of our DIDS data. Namely, DIDS has no effect on $(\Delta pH_S^*)_{NH_3}$ or P_f^* and yet blocks a substantial portion of $(\Delta pH_S^*)_{CO_2}$.

Action: We have revised the Discussion (see lines #853–863) to state the above. Recognizing that this line of logic requires a better presentation of the chemistry of pCMBS (see lines #890–897), we consulted an expert in mercury biochemistry (Leslie Poole) and introduced a new paragraph to the Discussion, at the beginning of the section “Effect of Inhibitors”.

Another important point that we had mentioned in the Discussion is that mathematical modeling (Somersalo et al, 2012) predicts that $(\Delta pH_S^*)_{CO_2}$ should increase linearly with the **log** of P_{M,CO_2} . This means that a 50% reduction of $(\Delta pH_S^*)_{CO_2}$ from each of pCMBS (and the same for DIDS) translates to >75% reduction of P_{M,CO_2} for each drug. This means that the pCMBS inhibition of its target spills over to the DIDS target (likely to be the CP) or that the inhibition by DIDS spills over to the monomeric pore. However, the DIDS spillover is highly unlikely because DIDS affects neither $(\Delta pH_S^*)_{NH_3}$ nor P_f^* . Thus, the most straightforward explanation is that DIDS blocks an alternate pathway (perhaps the central pore), reducing P_{M,CO_2} by >75% and that pCMBS blocks at least part of MP component of CO₂ permeability, which we suggest is <25%.

Action: We have revised the Discussion (see lines #933–945) to sharpen the log₁₀ discussion. In addition, we have sharpened the passage on spillover in the following paragraph (see lines #956–966).

[continuation of Referee's comment ...] Studies using mutated hAQP1 in which key residues necessary for monomeric pore transport have been mutated may be needed to conclude that the monomeric pore does not mediate 100% of human AQP1 CO₂ transport.

Response: It's complicated.

(a) In a paper in the late stages of preparation, we show that mutations and derivatization of the outer mouth of the central pore (CP) of human AQP5 markedly reduces CO₂ permeability without affecting P_f . The conclusion is that, for hAQP5, the vast majority of the CO₂ moves through the CP. However, the opposite is generally not true.

(b) In a study of naturally occurring SNPs of hAQP5, again in a paper nearly ready for submission, we show that a SNP (and other mutations to the same residue) at the selectivity filter of the MP—and this area is the logical target within the MP—greatly reduces both P_f and CO₂ permeability (even though in AQP5 the vast majority of CO₂ moves through the CP).

(c) In a study further away from submission, we have made other mutations at the selectivity filter of AQP5. Again, we see that such mutations markedly reduce P_f but also reduce CO₂ permeability much more than expected based on transit only through the MP.

(d) Thus, we suggest that pCMBS, which attacks C189 near the selectivity filter likely causes deformation of the CP. The reason, we believe, is that the selectivity filters of the 4 MPs back up against the 4 TM2 segments that line the extracellular half of the central pore. This would explain why pCMBS reduces P_{M,CO_2} by >75% when probably only <25% of the CO₂ moves through the MP per se.

(e) Thus, although the Reviewer's suggested mutations seem reasonable in isolation, our experience is that they are almost certain not to work. In summary, it seems relatively straightforward to block the CP without affecting the MP. It seems very difficult to block the MP without affecting the CP.

Action: The passage on spillover addresses the mechanism of spillover (see lines #956–966).

Minor comments:

1. The finding that pCMBS blocks only 50% of AQP1 CO₂ transport has been shown previously. This should be more explicitly stated in the manuscript.

Response: Thank you. We believe that the Referee is referring to the paper by Cooper (1998), which showed that pCMBS caused a statistically significant change (a decrease) in the maximal rate of pH_i decline— $(dpH_i/dt)_{Max}$ —during CO₂ addition, a significant difference that was present only in hAQP1-expressing oocytes but not in hAQP1-C189S or H₂O-injected oocytes. Most readers (including us) probably walked away from that paper thinking that pCMBS blocks all of the hAQP1-mediated CO₂ permeability. However, Cooper's statistical approach did not address that question. We believe that the newer approach of assessing $(\Delta pH_s^*)_{CO_2}$ is more precise than the earlier $(dpH_i/dt)_{Max}$ approach, making it easier to identify nuances in the effects of inhibitors.

We had referred to the Cooper paper several times but were remiss in not addressing the degree of blockade.

Action: We have extensively revised the Discussion (see lines #826–843) to address the above issues.

2. The anti-DIDS antibodies used in the studies do not appear to be affinity-purified. Is this correct? Can evidence of their specificity, using DIDS as a blocking antigen, be shown? Were comparable results obtained with antibodies from both animals?

Response: The anti-DIDS antibodies were affinity-purified using Protein A columns, which recognize the Fc region of the IgG antibodies. We did not affinity purify with the immunogen (KLH-DIDS) nor did we assess specificity with DIDS as a blocking antigen. However, we did evaluate the antigenicity towards DIDS-labeled AQP1 and observed similar banding patterns from the two rabbits among all six lots of anti-DIDS antibodies.

Action: The Methods (see lines #204–207) have been modified to address the concerns raised by Referee #1 and to also those of Reviewer #2 regarding rabbit husbandry during antibody production. We also edited the passage on antibody purification (see lines #522–524) to clarify our affinity purification and also the evaluation of antigenicity (see lines #528–530).

3. How many replicates of the data in Figure 2C were performed? Given that similar levels of expression of the two AQP1 proteins, wild-type and mutated, are essential for interpretation of the studies comparing transport of the 2 AQP1 proteins in this manuscript, can a summary figure showing the replicated data be added?

Response: We appreciate the referee's comment.

Action: We have now quantified the total expression of AQP1 wild-type and the C189S mutant in *Xenopus* oocytes in the 3 WB experiments, and modified the legend of Fig 2C to present these results. In addition, in response to the Editor’s comment, we replaced the photo in Fig 2C with a version showing a greater MW range.

4. What is the evidence that the effect of pCMBS is irreversible?

Response: In a search of the ms, we only can find the word “irreversible” used twice (see line #772 and line #930)—and “reversible” used nearby two other times—always in relation to DIDS. The evidence that DIDS blockade was irreversible under the conditions of the present study is the albumin wash in Fig 9A.

Action: For clarification, we added the word “subsequent” to the legend of Fig 9A.

5. Can the prior molecular dynamic simulation studies showing NH₃ movement through the monomeric AQP1 pore be discussed more explicitly?

Response: We thank the referee for this suggestion. The old text in the Introduction inappropriately included the reference to Assentoft, but did not include other references. We now reference the *AtTIP2;1* paper by Kirscht et al, which includes a model of NH₃ permeation. *Although the abstract of this paper says that they did experiments exploring the monomeric pore, a more careful reading of the paper shows that these were yeast growth-plate experiments, not biophysical experiments.*

Action: We now cite additional papers and, as the Referee suggested, include a summary of the Kirscht model (see lines #150-158). This required a modest rewrite of this passage.

Referee #2:

The current study provides a series of carefully performed and details experiments investigating the mechanisms by which water, ammonia and carbon dioxide are transported through AQP1. Based on selective inhibitors the data clearly provides evidence for two distinct pathways through AQP1 which possess different permeability properties. The monomeric AQP1 pores show permeability to all three substances while a DIDS selective pathway, likely through the central pore of the AQP1 tetramer allows passage of carbon dioxide but not water or ammonia. This work reinforces previous studies and inclusion of the C189S mutant data builds on the previous studies. The use of the anti-DIDS antibody provides some exciting insights into the mechanism of action of this drug, which appears to exert an action by crosslinking between the AQP1 monomers that form the tetramer.

Response: We sincerely thank the referee for the positive and insightful summary of our work.

General thoughts:

There are now several AQP1 modulators available which are specific for the AQP1 monomeric pore or block the cationic conductance. Have the authors investigated the actions of any of these compounds which would help support the conclusions of the current study.

Response: We are not sure to which AQP1 modulators the Referee refers. Verman et al (*Aquaporins: Important but elusive drug targets*, <https://doi.org/10.1038/nrd4226>) have summarized the difficulty in developing AQP-targeted drugs. We are aware of several reports of low-affinity inhibitors for which verification has been problematic. In the present paper, we used pCMBS because its precise target (C189) is well established.

While the authors have provided evidence that DIDS is crosslinking the AQP1 monomeric units the paper would be enhanced if they were able to provide a molecular basis for this action. It is well established that DIDS interacts with Lysine residues and a KXXX motif has been proposed as the basis of DIDS action in anion exchangers. Although AQP1 does not have such a motif, the K51 residue lines the tetrameric pore. Are the authors able to speculate on whether this would be a viable target for DIDS.

Response: We believe that there are 2 likely targets for the DIDS: K36 is in the middle of Loop A and K51 near the beginning of TM2 and the extracellular end of the tetrameric central pore. We agree that it would be extremely valuable to have structural data. Indeed, together with Ardi Vahedi-Faridi and Andreas Engle, we have begun a study of bovine RBCs derivatized with DIDS, and we have preliminary data on a single-particle structure of bovine AQP1 derivatized with DIDS.

Action: We have edited/expanded the Discussion to include a more careful consideration of DIDS chemistry (see lines #898-925) that underlies our work. We added a paragraph (see lines #918-925) on the likely targets of DIDS, with new insights from a colleague who is a structural biologist. Recognizing a disruption of flow, we moved the molecular dynamics section from this “DIDS” section to the end of the section titled “Pathways for NH₃ and CO₂” (now lines #878-888).

Paper review. Major comments:

The introduction provides a detailed overview of the literature linked to the transport of gases by Aquaporin. There is limited mention of the role of AQP1 as an ion channel and the potential

roles of the monomeric and/or central pores in this action. There is evidence to suggest that in these cases the central pore may be permeable to a substance (as mentioned in line 157) so it would be beneficial to expand the introduction to incorporate this information and put it in context with the current study. Additionally the Al-Samir (2025) study incorporates inhibitors which act on the ion conductance pathway, and it would be useful to address this in light of your current findings.

Response: Yasui and colleagues (1) demonstrated that the monomeric pore of AQP6 normally functions as an anion channel but that a single mutation—N60G in the middle of TM2, at a point where TM2 is lining the extracellular half of the central pore—abolishes anion permeability and effectively converts AQP6 to a water channel. Later we (2) made a similar mutation in hAQP5, finding that G50N eliminates both H₂O and CO₂ permeability. However, the nearby mutation L51R—the hydrophobic side chain of Leu-51 lines the central pore—converts AQP5 into an anion channel blockable by mercury. Thus, the current seems to flow through the monomeric pore. This L51R mutation is an example of how a single change can influence both the monomeric and central pores.

In addition, Yool and coworkers published several papers reporting that AQP1 can conduct cations. In response to the first Yool paper in Science in 1997, several laboratories published letters reporting that they could not reproduce the Yool findings.

As far as we are aware, the 2025 Al-Samir paper on RBCs does not address ion-conductance pathways.

Action: We have added to the Introduction a passage (see lines #106-116) in which we introduce the above material on AQP6, AQP5, and AQP1.

Overall the methods section feels very long for a paper of this nature. Although it is extremely detailed, in many places it is describing well established processes which have been described in depth previously. I feel the manuscript would benefit from the methods section being more concise.

Response: First, the protocol in the present ms is very complicated (hence, Fig 1). Second, in reviewing our previous experimental papers, we found that they were lacking a single, comprehensive, thorough listing of techniques in a way that an outsider could repeat the work a generation from now. Moreover, we were endeavoring to meet the more strenuous requirements of experimental description nowadays and felt it necessary to explain in full detail, including some steps that are previously well described due to complexity of many of the protocol maneuvers.

Line 464 - Pf assays. Can the authors confirm in this section how long the oocytes were in the hypotonic solution - is it 60s as suggested in the earlier methods. Based on previous studies using a similar construct, injection of oocytes with 25ng of cRNA is likely to create high expression of AQP1 protein with potential damage to the oocytes. Were the authors happy of the membrane integrity at the end of the first hypotonic exposure and how did they verify this.

Response: We confirm that the oocytes were exposed to hypotonic solution for 60 seconds during the P_f assays. Regarding the cRNA injection, we acknowledge that 25 ng is a relatively high amount, which can potentially stress the oocytes. However, the oocyte medium was changed daily, and only healthy oocytes with normal appearance were selected. Also, we performed experiments on Day 4, when the oocytes not only exhibited good expression but also were at their healthiest. During the experiments, membrane potential (V_m) was continuously monitored, and only oocytes exhibiting stable V_m (at least as negative as -40 as stated previously

on lines #381-382) and good morphology were included in the analysis. Furthermore, the same oocytes were used for electrophysiology and Pf assays at least twice, demonstrating their sustained viability.

Action: In the revised Methods section, we now specify the days on which we performed specific maneuvers on the oocytes (see lines #297-307). Also in Methods, we added a new inline heading (“Assessing oocyte health”, see lines #436-440). Finally, we added a line to indicate the timing of the P_f assay in hypotonic media (see lines #498-499).

Line 583 and Figure 2C. The authors have presented evidence that AQP1 and its C189S mutant are expressed at similar levels in oocytes. Can they please confirm if this was 3 preparations from different animals. Have the blots been quantified in any way to confirm similar levels of protein expression.

Response: Please see the response to Referee #1/Minor point #3, including ...

Action: ... modification of the legend of Fig 2C.

Line 592 and Figure 3. The authors indicate the pHS deflection is much smaller in control oocytes compared to oocytes expressing AQP1 or its C189s mutant. Have you confirmed this statistically? Given the other sections of the manuscript rely on the difference between the two values to give the channel dependent component of CO₂ transport it is crucial to state the significance.

Response: Thank you for raising this point, which we now have addressed.

Action: We have now added a new analysis of the “control” population data underlying Fig 3A, 3B and 3C (see lines #635-638 and #649-653), including a new “Statistics Table 3” which is also cross-referenced in the Figure 3 legend stating that “**Error! Reference source not found.** presents the analysis of the mean ΔpH_S magnitude recorded from all H₂O, AQP1 or AQP1-C189S “control” oocytes for which we show representative traces in panels A, B and C.”. Statistics Table 3 reports that the mean CO₂ elicited ΔpH_S transients recorded from AQP1 or AQP1-C189S expressing oocytes are both significantly larger than the H₂O oocytes but not significantly different from one another.

Line 660 and Figure 6 - See comments above for line 592 and Figure 3, the same idea applies.

Response: Thank you.

Action: Same as above. We have now added a new analysis of the “control” population data underlying Fig 6A, 6B and 6C (see lines #713-718) and included a new Statistics Table 6, and update the Figure 6 legend, citing Statistics Table 6.

Line 606-611 Could the authors include statistics to confirm that pCMBS has an effect on AQP1 but not the C189S mutant.

Response: This had already been done. Fig 4A shows that pCMBS reduces $(\Delta pH_S^*)_{CO_2}$, significantly for hAQP1-WT ($P=7.53 \times 10^{-6}$), whereas Fig 4B shows that the drug is without effect in hAQP1-C189S.

Line 613 - Can the authors include statistics to indicate the $(\Delta pH_S^*)_{CO_2}$ for AQP1 and C189S are very similar and not different from each other.

Response: This fell out of the ANOVA, 3 queries above.

Action: See the new Statistics table 3 (lines #635-638 and #649-653). A similar analysis applies to the NH₃ data, 2 queries above (see lines #713-718).

Minor comments:

Line 39 - Please check this doesn't make sense

Response: Thank you. Actually, it was good English, but perhaps too “literary”.

Action: We revised the sentence (see lines #39–40) to make it easier to read, still staying within the word limit.

Line 46 - after "by" please add a value

Response: Thank you.

Action: Done. The missing word was “half” (see line #46).

Line 89 - "of course" - not necessary

Response: Thank you. We were trying to show respect to Peter Agre.

Action: We have revised the sentence (see lines #89-90).

Page 7 - Comment 1 - not needed

Response: We presume that the Referee is referring to the footnote, which we included from internal identification of the antisera, which is available to other laboratories.

Action: We removed the footnote and put this explanation into the main text (see lines #202-203).

Line 185 - Please add details of Rabbit husbandry

Response: OK.

Action: We have updated the text to provide additional details on the rabbit husbandry when producing the polyclonal anti-DIDS antibody (see lines #204-207). In addition, we added an inline heading under “Generation of the DIDS antibody” (see line #519) and a new paragraph describing the evaluation of antigenicity (see lines #528-530).

Line 214 - Should "Solutions 4 and 5" be Solution 5 only. The description here is linked to the preparation of the 0.5mM NH₃/NH₄ solution

Response: We thank the referee for pointing this out. The description should indeed refer only to Solution 5. Solution 4 corresponds to the 5 mM NH₃/NH₄⁺ stock solution, and Solution 5 is the 0.5 mM NH₃/NH₄⁺ working solution obtained by a 1:10 dilution of Solution 4 into ND96.

Action: We have revised this passage to improve the clarity (see lines #238-239).

Line 564 The mechanism of the pH changes has been described in detail previously and this explanation could be shortened.

Response: We understand. However, our sense is that general audiences do not feel quite at home with the intricacies of the physiological chemistry of CO₂/HCO₃⁻ equilibria. Thus, we generally provide a bit more handholding than may be required in other subdisciplines.

Action: We have revised the text to shorten it somewhat (see lines #601-604).

Line 602 - The Preston et al, 1993 reference evaluates the effect of HgCl₂ on the C189S and does not address pCMBS. A different reference may be more appropriate here.

Response: We thank the referee for pointing out this potential source of confusion. Our understanding is that, like HgCl₂ (which can be quite toxic to living cells), pCMBS simply puts an “Hg” atom on the “S” of a Cys group. So, Preston et al deserve the credit for working this out.

Action: We moved the Preston reference so that it is immediately after the “HgCl₂” (see lines #646-647).

Line 671 - You indicate 11 oocytes in the text and 12 in Figure 6B - please check and confirm correct value

Response: Thank you for preventing that embarrassment!

Action: Corrected (see line #725).

Line 688 - The Macey (1984) study investigated the effect of mercurials on RBCs and not oocytes. Please consider if this reference is appropriate to be used here.

Response: Thanks for pointing out that bit of sloppy writing.

Action: We edited that passage (see line #742).

Literature cited

1. **Yasui M, Hazama A, Kwon TH, Nielsen S, Guggino WB, Agre P.** Rapid gating and anion permeability of an intracellular aquaporin. *Nature* 402: 184–187, 1999. doi: 10.1038/46045.
 2. **Qin X, Boron WF.** Mutation of a single amino acid converts the human water channel aquaporin 5 into an anion channel. *Am J Physiol Cell Physiol* 305: C663-672, 2013. doi: 10.1152/ajpcell.00129.2013.
-

END OF COMMENTS

The Physiological Society is a company limited by guarantee. Registered in England and Wales, No. 00323575. Registered Office: Hodgkin Huxley House, 30 Farringdon Lane, London, EC1R 3AW, UK. Registered Charity No. 211585. The Physiological Society and The Journal of Physiology are registered trademarks.

This email and any files transmitted with it are confidential and intended solely for the use of the individual or entity to whom they are addressed. If you have received this email in error please notify the sender. If you are not the named addressee you should not disseminate, distribute or copy this e-mail. The Physiological Society may monitor email traffic data.

The Physiological Society has taken reasonable precautions to ensure no viruses are present in this email, however does not accept responsibility for any loss or damage arising from the use of this email or attachments.

Dear Dr Musa-Aziz,

Re: JP-RP-2025-289556R1 "Mechanism of CO₂ and NH₃ Transport through Human Aquaporin 1: Evidence for Parallel CO₂ Pathways" by Raif Musa-Aziz, Robert Ryan Geyer, Seong-Ki Lee, Fraser J Moss, and Walter F Boron

Thank you for submitting your manuscript to The Journal of Physiology. It has been assessed by a Reviewing Editor and by 2 expert referees and we are pleased to tell you that it is acceptable for publication following satisfactory revision.

REVISION CHECKLIST:

We look forward to receiving your revised submission.

Yours sincerely,

Kim Barrett
Senior Editor
The Journal of Physiology

EDITOR COMMENTS

Reviewing Editor:

Thank you for addressing comments raised in previous review of your study, "Mechanism of CO₂ and NH₃ Transport through Human Aquaporin 1: Evidence for Parallel CO₂ Pathways".

Attached are critiques arising from review of this most recent version. Both Expert Referees appreciate the effort taken in addressing most of their concerns. The concern raised by Referee 1 regarding the extent of pCMBS's block remains a logical sticking point. Although Referee 1 offers that this non-experimentally validated assumption could be strengthened with support entailing an independent approach, this Referee also acknowledges that the conclusions nonetheless likely are valid. Overall, given the paucity of currently available tools, such testing is likely better suited for future studies.

Please also note that both Referees indicate several technical and presentation points that require your attention. I expect you will be able to correct readily.

REFEREE COMMENTS

Referee #1:

The manuscript "Mechanism of CO₂ and NH₃ transport through human aquaporin 1: evidence for parallel CO₂ pathways" by Musa-Aziz and colleagues uses physiologic approaches to assess the pathway(s) through which CO₂ and NH₃ are transported by hAQP1. The authors are experts in this field and in the techniques used in the manuscript.

My comments below follow the same organization as used in the original review.

Major comments

1. This concern dealt with whether pCMBS blocks 100% of CO₂ transport via the monomeric pore of hAQP1. If this is not the case, then the conclusion that there is a separate pathway for human AQP1 CO₂ transport not involving the monomeric pore is lost. The authors present a strong argument, albeit without direct experimental data, suggesting that it does, and this reviewer thinks they are likely correct.

Minor comments

1. Addressed

2. The term "affinity-purified" generally means that an antibody has been purified through affinity to the immunizing antigen. The use of a Protein A column that recognizes the Fc region of all IgG antibodies is not considered affinity purification. This very minor terminology discrepancy should be corrected. Also, the standard use of a new antibody involves showing that the immunizing antigen blocks immunoreactivity; this finding should be included in the manuscript.

3. Addressed.

4. Addressed.

5. Addressed.

Referee #2:

The authors have made significant alterations to the manuscript which addressed the issues raised previously.

A couple of very minor points to be checked in a final submission

Fig 1 - Panels D and J - The figure uses the same experimental trace for the pHs changes associated with NH_3/NH_4 . This is perfectly acceptable given it is to illustrate the technique, but the pH axis labels for the two traces are different - please review this.

Fig 9C and D in the legend mention effect of albumin washes, but for these panels the lack of impact of the Flag tag is being confirmed. Please check.

END OF COMMENTS

Dear Dr Musa-Aziz,

Re: JP-RP-2025-289556R1 "Mechanism of CO₂ and NH₃ Transport through Human Aquaporin 1: Evidence for Parallel CO₂ Pathways" by Raif Musa-Aziz, Robert Ryan Geyer, Seong-Ki Lee, Fraser J Moss, and Walter F Boron

Thank you for submitting your manuscript to The Journal of Physiology. It has been assessed by a Reviewing Editor and by 2 expert referees and we are pleased to tell you that it is acceptable for publication following satisfactory revision.

Your revised manuscript should be submitted online using the link in your Author Tasks <https://jp.msubmit.net/cgi-bin/main.plex?el=A4JS5HWQ2B7Vje7F4A9ftdApyMXYmaP5Q0mZ5iaARxQZ>.

This link is accessible via your account as Corresponding Author; it is not available to your co-authors. If this presents a problem, please contact journal staff (jp@physoc.org). Image files from the previous version are retained on the system. Please ensure you replace or remove any files that are being revised.

LANGUAGE EDITING AND SUPPORT FOR PUBLICATION: If you would like help with English language editing, or other article preparation support, Wiley Editing Services offers expert help, including English Language Editing, as well as translation, manuscript formatting, and figure formatting at www.wileyauthors.com/eo/preparation. You can also find resources for Preparing Your Article for general guidance about writing and preparing your manuscript at www.wileyauthors.com/eo/prepresources.

REVISION CHECKLIST:

We look forward to receiving your revised submission.

Yours sincerely,

Kim Barrett

Senior Editor

The Journal of Physiology

EDITOR COMMENTS

Reviewing Editor:

Thank you for addressing comments raised in previous review of your study, "Mechanism of CO₂ and NH₃ Transport through Human Aquaporin 1: Evidence for Parallel CO₂ Pathways".

Attached are critiques arising from review of this most recent version. Both Expert Referees appreciate the effort taken in addressing most of their concerns. The concern raised by Referee 1 regarding the extent of pCMBS's block remains a logical sticking point. Although Referee 1 offers that this non-experimentally validated assumption could be strengthened with support entailing an independent approach, this Referee also acknowledges that the conclusions nonetheless likely are valid. Overall, given the paucity of currently available tools, such testing is likely better suited for future studies.

Please also note that both Referees indicate several technical and presentation points that require your attention. I expect you will be able to correct readily.

REFEREE COMMENTS

Referee #1:

The manuscript "Mechanism of CO₂ and NH₃ transport through human aquaporin 1: evidence for parallel CO₂ pathways" by Musa-Aziz and colleagues uses physiologic approaches to assess the pathway(s) through which CO₂ and NH₃ are transported by hAQP1. The authors are experts in this field and in the techniques used in the manuscript.

My comments below follow the same organization as used in the original review.

Major comments

1. This concern dealt with whether pCMBS blocks 100% of CO₂ transport via the monomeric pore of hAQP1. If this is not the case, then the conclusion that there is a separate pathway for human AQP1 CO₂ transport not involving the monomeric pore is lost. The authors present a strong argument, albeit without direct experimental data, suggesting that it does, and this reviewer thinks they are likely correct.

Response: Thank you for your comments. This dialogue and response made the paper stronger.

Minor comments

1. Addressed. **Response:** Thank you

2. The term "affinity-purified" generally means that an antibody has been purified through affinity to the immunizing antigen. The use of a Protein A column that recognizes the Fc region of all IgG antibodies is not considered affinity purification. This very minor terminology discrepancy should be corrected.

Response: Thank you for highlighting this important distinction. We confirm that the antibody was purified using a Protein A column, which does not constitute affinity purification in the strict sense. We have corrected the terminology accordingly.

Action: On line 522 we deleted the word “affinity”

Also, the standard use of a new antibody involves showing that the immunizing antigen blocks immunoreactivity; this finding should be included in the manuscript.

Response: We lack a blot demonstrating that peptide pre-absorption blocks immunoreactivity, but Figure 10D shows that spheroplast preps **not** treated with DIDS (Figure 10D, lanes 1, 3, and 5), which contain AQP1 protein (as shown in the same lanes of Figure 10C probed with anti-Flag antibody), have no detectable immunoreactivity vs. the anti-DIDS antibody.

Action: We have made the following revision to the Figure 10D legend (underlined text is new):
“Western blot of material from the same experiment as depicted in panels B and C, probed with anti-DIDS. The lane assignments are the same as in panel C. The results in panels A through D are representative of two independent experiments. The absence of immunofluorescence in lanes 1, 3 and 5, especially where there is corresponding immunoreactivity for FLAG-tagged hAQP1-WT in panel C, demonstrates the specificity of the anti-DIDS antibody”.

3. Addressed. **Response:** Thank you.

4. Addressed. **Response:** Thank you.

5. Addressed. **Response:** Thank you.

Referee #2:

The authors have made significant alterations to the manuscript which addressed the issues raised previously.

A couple of very minor points to be checked in a final submission

Fig 1 - Panels D and J - The figure uses the same experimental trace for the pH_s changes associated with NH₃/NH₄⁺. This is perfectly acceptable given it is to illustrate the technique, but the pH axis labels for the two traces are different - please review this.

Response: Thanks for spotting that embarrassing mistake.

Action: The label on scale for the 0.5 mM NH₄⁺/NH₃ trace in Figure 1D has been amended to be from pH 7.3 -7.5 as in Figure 1J.

Fig 9C and D in the legend mention effect of albumin washes, but for these panels the lack of impact of the Flag tag is being confirmed. Please check.

Response: Thanks again for spotting that embarrassing mistake

Action: We deleted “effect of albumin washes on” from the legend for both Panels C and D.

END OF COMMENTS

The Physiological Society is a company limited by guarantee. Registered in England and Wales, No. 00323575. Registered Office: Hodgkin Huxley House, 30 Farringdon Lane, London, EC1R 3AW, UK. Registered Charity No. 211585.

The Physiological Society and The Journal of Physiology are registered trademarks.

This email and any files transmitted with it are confidential and intended solely for the use of the individual or entity to whom they are addressed. If you have received this email in error please notify the sender. If you are not the named addressee you should not disseminate, distribute or copy this e-mail. The Physiological Society may monitor email traffic data.

The Physiological Society has taken reasonable precautions to ensure no viruses are present in this email, however does not accept responsibility for any loss or damage arising from the use of this email or attachments.

Dear Dr Musa-Aziz,

Re: JP-RP-2025-289556R2 "Mechanism of CO₂ and NH₃ Transport through Human Aquaporin 1: Evidence for Parallel CO₂ Pathways" by Raif Musa-Aziz, Robert Ryan Geyer, Seong-Ki Lee, Fraser J Moss, and Walter F Boron

We are pleased to tell you that your paper has been accepted for publication in The Journal of Physiology.

Yours sincerely,

Kim Barrett
Senior Editor
The Journal of Physiology

IMPORTANT POINTS TO NOTE FOLLOWING ACCEPTANCE OF YOUR PAPER:

- You can help your research get the attention it deserves! Check out Wiley's free Promotion Guide for best-practice recommendations for promoting your work at: www.wileyauthors.com/eeo/guide. You can learn more about Wiley Editing Services which offers professional video, design, and writing services to create shareable video abstracts, infographics, conference posters, lay summaries, and research news stories for your research at: www.wileyauthors.com/eeo/promotion.

- If you would like to receive our 'Research Roundup', a monthly newsletter highlighting the cutting-edge research published in The Physiological Society's family of journals (The Journal of Physiology, Experimental Physiology, Physiological Reports, The Journal of Nutritional Physiology and The Journal of Precision Medicine: Health and Disease), please click this link, fill in your name and email address and select 'Research Roundup': <https://www.physoc.org/journals-and-media/membernews>

EDITOR COMMENTS

Reviewing Editor:

I appreciate the Authors' response to comments arising from the last round of review.

Just a few notes:

The response regarding Referee 1's distinction of affinity purification refers to the line number incorrectly; "affinity" appears to be removed from line 532, not line 522. Line 522 appears in the preceding section pertaining to the generation of the DIDS fusion protein. In any case, the adjustment effectively corrects the matter.

The response to Referee 1's comment regarding the best practice of furnishing an antigen-blocked control for the new anti-DIDS antibody is not perfect. However, it seems the Authors lack such a control. However, the Authors point out an internal experimental control in figure 10 D. They incorporate this information the accompanying legend. I believe this acceptably addresses the Referee's concern, if not in the most formal sense.

Corrections requested by Referee 2 are addressed satisfactorily.